# Modelling the biogeochemical effects of heterotrophic and autotrophic N₂ fixation in the Gulf of Aqaba (Israel), Red Sea

Angela M. Kuhn[1, 2], Katja Fennel[1], Ilana Berman-Frank[3, 4]

[1]Department of Oceanography, Dalhousie University, Halifax, B3H 4R2, Canada
[2]Scripps Institution of Oceanography, University of California San Diego, La Jolla, 92093-0021, USA*
[3]Mina and Everard Goodman Faculty of Life Sciences, Bar Ilan University, Ramat Gan, 5290002, Israel
[4]Leon H. Charney School of Marine Sciences, University of Haifa, Mt. Carmel, Haifa, 3498838, Israel*

*current address

*Correspondence to*: Angela M. Kuhn (angela.kuhn@dal.ca)

**Abstract.** Recent studies demonstrate that marine N₂ fixation can be carried out without light by heterotrophic N₂-fixers (diazotrophs). However, direct measurements of N₂ fixation in aphotic environments are relatively scarce. Heterotrophic, as well as unicellular and colonial photoautotrophic diazotrophs, are present in the oligotrophic Gulf of Aqaba (northern Red Sea). This study evaluates the relative importance of these different diazotrophs by combining biogeochemical models with
time series measurements at a 700m-deep monitoring station in the Gulf of Aqaba. At this location, an excess of nitrate, relative to phosphate, is present throughout most of the water column and especially in deep waters during stratified conditions. A relative excess of phosphate occurs only at the water surface during nutrient-starved conditions in summer. We show that a model without N₂ fixation can replicate the observed surface chlorophyll but fails to accurately simulate inorganic nutrient concentrations throughout the water column. Models with N₂ fixation improve simulated deep nitrate by enriching sinking
organic matter in nitrogen, suggesting that N₂ fixation is necessary to explain the observations. The observed vertical structure of nutrient ratios and oxygen is reproduced best with a model that includes heterotrophic, colonial and unicellular autotrophic diazotrophs. These results suggest that heterotrophic N₂ fixation contributes to the observed excess nitrogen in deep water at this location. If heterotrophic diazotrophs are generally present in oligotrophic ocean regions, their consideration would increase current estimates of global N₂ fixation and may require explicit representation in large-scale models.

## 1 Introduction

Biological nitrogen fixation refers to the conversion of dinitrogen gas (N₂) via reduction to ammonium (NH₄) into bioavailable forms of nitrogen by a specialized group of microbes containing the nitrogenase enzyme complex. On geological timescales, the size of the oceanic reservoir of bioavailable nitrogen, and thus the ocean's capacity for exporting carbon to depth, is controlled by the balance between removal of fixed nitrogen by denitrification and input by N₂ fixation (Falkowski, 1997;
Haug et al., 1998; Deutsch et al., 2007; Gruber and Galloway, 2008; Fennel et al., 2005). The amount of organic matter exported from the surface to the deep ocean (i.e., export production) depends on allochthonous inputs of nitrogen (i.e., "new

nitrogen") into the euphotic zone (Eppley and Peterson, 1979). These new nitrogen inputs determine the amount of "new production", which is directly related to the exported fraction. Locally the supply of new nitrogen can occur through several mechanisms, including microbially mediated $N_2$ fixation, diapycnal mixing injecting deep nitrate ($NO_3$) into the surface, lateral transport, atmospheric sources and riverine input. While the injection of deep $NO_3$ is often regarded as the dominant source of
new nitrogen that drives the seasonal cycle of marine primary production; there is significant interest in quantifying the contribution of $N_2$ fixation to primary production, particularly in oligotrophic areas (Karl, 2002; Zehr and Ward, 2002; Capone et al., 2005; Luo et al., 2012).

Diazotrophs contain nitrogenase, the catalyst enzyme for $N_2$ reduction, which is encoded by *nif* genes. *Trichodesmium spp.,* a group of non-heterocystous filamentous cyanobacteria that form large colonies, were traditionally considered the main
contributors to $N_2$ fixation in the surface subtropical and tropical ocean (Carpenter and McCarthy, 1975; Capone et al., 2005). Increased sampling efforts and method improvements subsequently led to the discovery of diverse diazotroph groups including heterocystous endosymbiotic cyanobacteria (Zehr et al., 1998; Carpenter et al., 1999), free-living unicellular cyanobacteria (Zehr et al., 2001; Montoya, 2004; Moisander et al., 2010), and other cyanobacterial symbionts (Zehr et al., 2000). Most recently, genetic techniques have allowed the detection of *nif* genes in a number of anaerobic and heterotrophic phylotypes
(Zehr et al., 2008; Zehr, 2011; Rahav et al., 2013, 2015). The abundance of *nif* genes does not necessarily imply that these organisms are actively fixing $N_2$ (Zehr et al., 2000; Moisander et al., 2017); however, the correlation between heterotrophic bacterial productivity and $N_2$ fixation rates suggests that significant aphotic $N_2$ fixation may occur in the Red Sea (Rahav et al., 2013, 2015, Benavides et al., 2018). The differences in size and physiology of these diverse diazotrophs also suggest that they occupy distinct niches, and thus may affect primary productivity and export production differently (Bonnet et al., 2016;
Moisander et al., 2010).

Most biogeochemical models treat $N_2$ fixation as a purely light-dependent, autotrophic process. These models either use standard formulations of light limitation for the diazotrophic groups (e.g., Fennel et al., 2002; Moore, et al., 2004; Gregg, 2008; Dutkiewicz et al., 2012), or include theoretical considerations to introduce an empirical $N_2$ fixation flux in the model (e.g., Bisset et al., 1999). Some approaches neglect light limitation on diazotrophy and instead infer global $N_2$ fixation patterns from
the distribution of dissolved inorganic nitrogen and phosphorus, and estimates of ocean circulation (e.g., Deutsch et al., 2007). In trait-based models, diazotrophs are usually accounted for by a single functional group with biological rate parameters intended to represent either *Trichodesmium spp.*, unicellular cyanobacteria, or a generic autotrophic diazotroph. More complex models of diazotroph growth have explored cellular function and the effects of a variable cellular stoichiometry, however still limited by light (Kreuss et al., 2015; Fernández-Castro et al., 2016). Only a few modelling studies have evaluated multiple
autotrophic diazotroph groups simultaneously, by considering separate groups for *Trichodesmium spp.*, unicellular cyanobacteria and diatoms-cyanobacterial associations (e.g., Monteiro et al., 2010; Dutkiewicz et al., 2012, 2015). To our knowledge, heterotrophic $N_2$ fixation has not yet been considered explicitly in biogeochemical models.

Understanding the ecological dynamics of different types of diazotrophs should significantly improve predictive capabilities in biogeochemical models, and lead to more accurate estimates of global $N_2$ fixation rates. It has been suggested that $N_2$ fixation rates are underestimated globally due to limited knowledge about the distribution and characteristics of $N_2$ fixing organisms (Montoya, 2004; Zehr, 2011). It is also assumed that marine $N_2$ fixation may increase globally, as a result of ocean warming and higher concentrations of dissolved $CO_2$ in sea water (Hutchins et al., 2007; Levitan et al., 2007; Dutkiewicz et al., 2015). To our knowledge, these laboratory experiments have only explored the response of *Trichodesmium*. Less information is available about the effects of climate trends on other diazotrophic organisms.

In this study we explore the biogeochemical signatures that result from different assumptions about the ecological niches occupied by diazotrophs. Our study area is the Gulf of Aqaba, a northern extension of the Red Sea. Aside from the reported presence of diverse diazotroph types, the morphology of the Gulf of Aqaba limits horizontal transport of deep waters, thus allowing us to simplify the physical model's complexity and focus on the biological component. We aim to answer the following two questions: i) How important is $N_2$ fixation as a source of new nitrogen in the Gulf of Aqaba? ii) How important is heterotrophic, light-independent $N_2$ fixation? To address these questions, we implemented a one-dimensional model at a monitoring station for which monthly quality-controlled measurements of physical and biogeochemical variables are available from 2004 onward. We then systematically tested different model assumptions about diazotrophy and calibrated selected model parameters to facilitate an objective comparison between the different biogeochemical model versions. The different assumptions about diazotrophy consider the characteristics of organisms identified in the Gulf of Aqaba, including heterotrophic fixing α and γ proteobacteria (Rahav et al., 2013), unicellular cyanobacteria, and *Trichodesmium spp.* (Post et al., 2002; Foster et al., 2009; Rubin et al., 2011). Our most important conclusion is that aphotic $N_2$ fixation is necessary to reproduce the observed excess nitrogen in deep waters of the Gulf, while maintaining reasonable surface $N_2$ fixation rates. Our best performing biogeochemical model of the Gulf of Aqaba estimates annual $N_2$ fixation rates in overall agreement with local and large-scale estimates in the literature.

## 2 Study Area: The Gulf of Aqaba

The Gulf of Aqaba is a quasi-rectangular, 200-km long, 20-km wide, semi-enclosed basin in the northeast region of the Red Sea (Figure 1). The average depth of the Gulf of Aqaba is 800 m, its deepest point approximately 1800 m, and it is surrounded by arid mountains that steer the dominantly northerly winds (Berman et al., 2003). Two shallow sills, the Bab el Mandeb (~140 m) and the Strait of Tiran (~240 m), inhibit the entrance of cold and dense deep waters from the Indian Ocean. Since inflow is restricted to warm surface waters, the Gulf's deep-water masses (>300 m) are locally formed (Wolf-Vecht et al., 1992; Biton et al., 2008) and have negligible horizontal transport toward the exterior (Klinker et al., 1976; Manasrah et al., 2006).

The annual hydrographic cycle exhibits a well-defined seasonality where vertical temperature and salinity distributions are dominantly affected by surface heat fluxes and modified by surface advective fluxes (Carlson et al., 2014). During winter

(December to March), convective vertical mixing usually extends to depths >300 m (Labiosa et al., 2003), and even reaches the bottom (~700-800 m bottom depth) in some extreme years (Figure 2, 3). From April to September, the water column is thermally stratified, and inflowing warm surface waters from outside the Gulf occupy the layer above the thermocline (Genin and Paldor, 1998; Berman et al., 2000; Biton and Gildor 2011). During fall (October to December), surface cooling and high evaporation rates erode the seasonal stratification and re-establish a well-mixed water column (Berman et al., 2003; Monismith and Genin, 2004). The Gulf experiences net evaporation of approximately 1.6 m $yr^{-1}$ (Ben-Sasson et al., 2009) due to negligible precipitation and run-off (Wolf-Vecht et al., 1992).

The Gulf is oligotrophic, with surface nitrate ($NO_3$) and phosphate ($PO_4$) concentrations usually close to their detection limits during summer stratification (Fuller et al., 2005; Mackey et al., 2009; Meeder et al., 2012). Deep winter mixing supplies inorganic nutrients to the surface, and $NO_3$ and $PO_4$ reach ~2 µM and ~0.1 µM, respectively (Figure 2; Lindell and Post 1995; Lazar et al. 2008). Dust from the desert provides a sufficient atmospheric source of soluble iron (Fe) for microbial growth in the Gulf (Chase et al., 2006; Chen et al., 2007). Phytoplankton spring blooms reach a maximum chlorophyll concentration of ~2 mg Chl-a $m^{-3}$ and their initiation is strongly correlated with the termination of winter cooling of the water column (Zarubin et al., 2017). Interannual variability in the depth of winter convective mixing results in periods of nutrient accumulation in deep waters (Figure 2; Wolf-Vecht et al., 1992; Lazar et al., 2008; Carlson et al., 2012), which is re-set during extreme winter mixing events approximately every four years (Silverman and Gildor, 2008). The periodicity of these extreme mixing events has been associated with regional weather patterns that modify the Red Sea water temperatures (Silverman and Gildor, 2008).

## 3 Methods

We analysed the role of autotrophic and heterotrophic $N_2$-fixing organisms in determining biogeochemical patterns at an open pelagic site (Station A), located in the northern Gulf of Aqaba, by testing four main alternative ecosystem model versions and six model subversions with minor variations. The ecosystem models are evaluated in terms of their ability to replicate observations of oxygen ($O_2$), $NO_3$, $PO_4$, and chlorophyll. In this section, we first describe the available observations, then the models, and finally the systematic model calibration method.

### 3.1 Observations

Meteorological and oceanographic observations are available from the Inter-University Institute (IUI) for Marine Sciences in Eilat, Israel (http://www.iui-eilat.ac.il/Research/ NMPMeteoData.aspx). Meteorological observations are used to calculate surface heat and momentum fluxes for the physical model and incoming light for the biological models. Observed meteorological variables include wind speed, air temperature, air humidity, air pressure, irradiance and cloud cover. This data has been collected continuously and automatically at 10 min intervals by the meteorological instrumental array at the end of the IUI pier since 2006.

Monthly CTD and bio-chemical profiles at Station A (29.5° N, 34.9° E) were collected during monthly surveys of the National Monitoring Program (NMP) from 2004 to 2014 (http://www.iui-eilat.ac.il/Research/NMPMeteoData.aspx). CTD profiles are used to nudge temperature and salinity in the physical model (see section 3.2). Bio-chemical profiles, including $NO_3$, nitrite ($NO_2$), ammonium ($NH_4$), $PO_4$, $O_2$ and chlorophyll-a (Chl-a), are used for biogeochemical model calibration (years 2006 to 2010) and model validation against unassimilated data (years 2011 to 2014). Nutrients were measured using spectrophotometry (QuickChem 8000 flow injection), $O_2$ was determined by Winkler titrations, and Chl-a concentrations are estimated using fluorometry (Turner Designs 10-AU).

### 3.2 Model Descriptions

The ecosystem models are implemented within the General Ocean Turbulence Model (GOTM), a one-dimensional physical model that computes solutions to differential equations for the vertical transport of momentum, salt and heat (Burchard et al., 1999). GOTM is implemented for the 700-m deep station with a vertical resolution of 3 m and forced with hourly meteorological observations from the IUI pier. Temperature and salinity are nudged to observed CTD profiles with a nudging time scale of 30 days. This is done to account for the influence of horizontal advection of heat and salt in the one-dimensional model and ensures a realistic representation of density stratification. The effect of temperature and salinity nudging on the results is analysed below (section 4.2.1). As model calibration is computationally expensive, model simulations run only from January 2005 to September 2010. The first year of each simulation is considered model spin-up and excluded from further analysis (climatological meteorological forcing is used for the first year, as this database starts in 2006). $NO_3$, $NH_4$, $PO_4$ and $O_2$ initial total concentrations match the observed total inventories, using homogenous concentrations throughout the water column of 2.5 mmol m$^{-3}$, 0.05 mmol m$^{-3}$ and 0.15 mmol m$^{-3}$, respectively. Vertical nutrient and oxygen concentrations are redistributed within a few months and replicate the observed vertical distributions well starting from October 2005. Non-$N_2$-fixing phytoplankton, zooplankton and detritus are initially set to a homogeneous small value of 0.1 mmol N m$^{-3}$, and also readjust rapidly because the adjustment timescales for these variables are short (Fennel et al., 2006). Diazotrophs initial values are set to lower densities than non-$N_2$-fixing organisms, with a homogenous total value of 0.03 mmol N m$^{-3}$ (i.e., models with multiple diazotrophs maintain the same amount of initial diazotrophic biomass by dividing initial values equally among organisms).

Four main ecosystem model versions of increasing complexity (referred to as H0, H1, H2 and H3) are treated as alternative hypotheses of how biological processes, especially diazotrophy, control the vertical distribution and temporal variability of dissolved inorganic nutrients and oxygen. H0 is the base model without diazotrophs and follows the model equations described in Fennel et al., (2006, 2013). In general, the model follows Monod kinetics using a fixed N:P ratio ($R_{N:P}^{nf} = 16$). Sensitivity to the constant N:P ratio is explored in section 4.2.3. We test the H0 model with and without a sediment denitrification flux (model versions H0 and H0', respectively). H0 includes denitrification but no $N_2$ fixation, as no diazotrophs are considered. H0' does include neither denitrification, nor diazotrophic organisms, and thus the underlying assumption of this model version

is that there is a balance between inputs from $N_2$ fixation and losses of fixed nitrogen due to denitrification. When present, the denitrification flux follows Fennel et al. (2013) with a loss fraction of 6 mol $N_2$ per mol of organic matter remineralized at the sediment-water interface. This generates an average sediment denitrification flux of $0.25 \pm 0.46$ mmol N m$^{-2}$ d$^{-1}$, with a maximum value of 3.01 mmol N m$^{-2}$ d$^{-1}$.

H1, H2 and H3 are modified versions of H0, in which different groups of diazotrophic organisms are added sequentially. Diazotrophic growth formulations are similar to those of non-$N_2$-fixing phytoplankton, except that they are not limited by nitrogen and have a higher N:P ratio ($R^f_{N:P} = 45$). H1 introduces a generic autotrophic diazotroph; H2 replaces H1's generic diazotroph with two autotrophs representing unicellular and colonial (e.g., *Trichodesmium spp.*) cyanobacteria. The unicellular group overall follows the same formulation as the generic diazotrophs, except that no aggregation term is included. We simplify
unicellular diazotroph behaviour in this manner because this group represents free-living picoplanktonic cells that typically do not form large colonies although they can aggregate (Bonnet et al. 2016). Instead, we assume this group is grazed by zooplankton at similar rates as non-$N_2$-fixing phytoplankton. This difference between colonial and unicellular groups is consistent with studies suggesting that colonies represent an evolutionary adaptation to decrease grazing pressure (Nielsen, 2006). Aside from their size, *Trichodesmium spp.* colonies may be less palatable and harder to digest due to toxins (Kerbrat et
al. 2010, 2011). Grazing is not a major fate of this group (O'Neil and Roman, 1994).

The last model version, H3, adds a heterotrophic group to the model structure of H2. This functional group is not limited by light and grows by consuming both dissolved inorganic and organic forms of nutrients. An intermediate version H3' is used as a control, where the heterotrophic organisms do not fix nitrogen and are limited by the availability of nitrogen in inorganic forms and from small detritus. Model H3 eliminates the nitrogen limitation and the heterotrophic group becomes a
heterotrophic diazotroph group. In a subsequent set of four model experiments (H3a, H3b, H3c, and H3d), we remove complexity from H3. The heterotrophic group remains, but we sequentially remove the autotrophic groups one-at-a-time: first the colonial cyanobacteria (H3a), then the unicellular cyanobacteria (H3b), then the generic autotrophic diazotroph (H3c), and finally we remove all autotrophic diazotrophs (H3d). A summary of all model versions is given in Table 1 and a description of state variables and full model equations in the Supplement.

We acknowledge that some of these model assumptions are still simplifications of diazotroph behaviour. For example, *Trichodesmium spp.*, as well as other autotrophic and diazotrophic organisms, may also take up dissolved organic matter to support growth (Benavides et al., 2017). Nevertheless, model assumptions and formulations used here are in line with the most commonly accepted understanding of the dominant controls on microbial and diazotrophic growth.

### 3.3 Model Parameters

### 3.3.1 Parameter Optimization Method

Parameter optimization refers to the minimization of misfit between model and observations by adjusting model parameters. We applied the method first to systematically calibrate the most sensitive parameters of H0 (see supplement), and then to independently re-calibrate parameters in H1 to H3 after the introduction of diazotrophs. We used an evolutionary algorithm, where changes in the parameter values follow a set of rules inspired by the process of natural selection (Houck et al., 1995; Kuhn et al., 2015). The algorithm starts with a randomly generated "population" of 30 parameter sets $(\vec{p})$, which are iteratively modified over a number of generations. During each generation of the population, the cost $J(\vec{p})$ of the model with parameter set $\vec{p}$ is calculated as:

$$J(\vec{p}) = \frac{1}{V}\sum_{v=1}^{V}\frac{w_v}{N}\sum_{i=1}^{N}\left(\hat{y}_{v,i} - y_{v,i}\right)^2,\tag{1}$$

where $\hat{y}$ represents a model estimate and $y$ the corresponding observation. $N$ is the number of observations included for each variable $v$. Here the number of variables $V$ is 5 (nitrate + nitrite, ammonium, phosphate, chlorophyll-a, and oxygen measured as profiles at Station A between 2006 and 2010). Model-data misfits are weighted by the factor $w_v = 1/\sigma_v$, i.e, the inverse standard deviation of each variable. Half of the parameter sets with the lowest $J$ value "survive" to the next generation. The other half of the population is regenerated from new parameter sets obtained by recombination of two random "parent" sets drawn from the better performing half (i.e., the "survivors" of the previous generation). Parameters also "mutate", i.e. random noise is added, for additional variability in the parameter space. An allowable range of values is set for each parameter based on the literature (Table 2).

### 3.3.1 Optimized Parameters

The parameter optimization method has limitations. Most importantly, the optimization cannot estimate with confidence parameters that are unconstrained by the observations (Fennel et al. 2001; Schartau and Oschlies, 2003; Ward et al., 2010). To avoid this, a subset of H0's most sensitive parameters was selected for optimization through a preliminary sensitivity analysis. Optimized parameters for H0 are identified in Table 2 along with the optimal values. The optimization was replicated 10 times over 100 generations using the algorithm described in section 3.3.1. Non-optimized parameters are fixed at their a priori estimates based on Fennel et al. (2006, 2013).

For each model version with diazotrophs (H1, H2 and H3), some of the parameters already optimized for H0 required re-calibration to properly accommodate the changes in system dynamics. Re-calibrated parameters for each model version are presented in Table 3. No re-calibration was performed for model versions (H0' and H3a-d), as they are aimed to test the relative importance of individual model components.

### 3.3.2 Diazotroph Parameters

Since none of the parameters directly related to the diazotroph groups are constrained by the available observations, they were predefined for H1, H2 and H3, based on the observational and modelling literature (Table 3). Previous modelling studies have used maximum growth rates of generic $N_2$ fixers ranging from 0.4 $d^{-1}$ (Moore, et al., 2004) to 1.25 $d^{-1}$ (Ward et al., 2013). When model diazotrophs are assumed to represent *Trichodesmium spp*. values range between 0.17 $d^{-1}$ (Hood et al., 2001) to 0.3 $d^{-1}$ (Fennel et al., 2002). From the observational literature, *Cyanothece* (unicellular cyanobacteria) and *Trichodesmium spp*. cultured under various combinations of Fe and light availability exhibit maximum rates around $0.3 \pm 0.05$ $d^{-1}$ (Capone et al., 1997; Berman-Frank et al., 2001; Hutchins et al., 2007). Growth rates can be higher (up to 0.5 $d^{-1}$) at high $CO_2$ and high light availability (Kranz et al., 2010; Hong et al., 2017). We chose a common reference maximum growth rate of 0.25 $d^{-1}$ for all photosynthetic diazotrophs, such that differences between the model versions result only from the different assumptions about the losses of each group (e.g., predation of unicellular cyanobacteria vs. sinking of large aggregates). Based on growth rates measured for cultured heterotrophic bacteria, we chose a value of 0.2 $d^{-1}$ for the heterotrophic diazotrophs (Pomeroy and Wiebe, 2001). Observational and modelling studies were also considered to set the photosynthetic initial slope of photosynthetic diazotrophs (Geider et al., 1997; Moore et al., 2004; Hutchins et al., 2007). Other parameters are based on Fennel et al. (2002).

### 4 Results

### 4.1 Observed NO$_3$ and PO$_4$ Patterns

To provide context for the evaluation of our model simulations, we first describe the observed interannual and seasonal variability of $NO_3$ and $PO_4$ for the complete time series (2004 to 2014) at Station A (Figure 2). From May to January vertical distributions of $NO_3$ and $PO_4$ show depletion of nutrients in the euphotic zone and a nutricline between 100 and 200 m. From February to April, nutrient concentrations increase near the surface and decrease in deep waters (>200 m) as result of vertical mixing. Multi-year periods of accumulation of nutrients in deep waters were observed from: i) the beginning of the series to the end of 2006, ii) after the winter of 2008 until February 2012, and iii) after the winter of 2013 until the end of the series. These periods are bookended by winters with extremely deep mixing events in 2007, 2008, 2012 and 2013 during which nutrient concentrations are nearly homogenized in the entire water column. Two prolonged periods of these vertically homogenous conditions were observed in 2007 and 2008, lasting two to three months.

Our model calibration simulations are from 2006 to 2010, allowing us to include two years with deep winter mixing (2007 and 2008) and two years with moderate winter mixing (2009 and 2010). Figure 3 shows the linear metric N* (N*=DIN-16DIP) for our simulation period which quantifies excess and deficit of nitrogen relative to phosphorus with respect to the canonical Redfield ratio (N:P=16:1). This metric thus allows diagnosing patterns of net nitrogen addition, i.e. the balance of $N_2$ fixation and denitrification, on global and local scales (e.g., Gruber and Sarmiento, 1997). Using the Redfield ratio as a reference, N* is insensitive to changes in nutrient concentrations that result from nutrient uptake by non-$N_2$-fixing phytoplankton and

remineralization of organic matter, assuming these processes occur in Redfield stoichiometry. Positive N* reflects an excess of nitrogen (DIN) and can be interpreted as a signature of $N_2$ fixation.

N* values presented in Figure 3 are calculated using the observations from Figure 2. Excess nitrogen (between +0.20 and +1.06 mmol N $m^{-3}$) dominates throughout most of the water column, except at the surface during stratified summer conditions,
when nutrients are depleted and surface waters exhibit an excess of phosphate (-0.35± 0.25 mmol N $m^{-3}$). Waters with excess nitrate are brought to the surface during winter; however, N* values rapidly return to negative at the surface. The magnitude and duration of positive surface N* values appear to be related to the depth of winter mixing.

### 4.2 Model Results

### 4.2.1 Sensitivity to Physical Nudging

Model runs, with and without temperature and salinity nudging towards observations, demonstrate that nudging has a negligible effect below 200 m, indicating that horizontal advection does not modify the lower part of the water column in a significant way (supplement). Above 200 m, nudging corrected model errors in the representation of vertical mixing and surface forcing. The average magnitude of the differences due to nudging in the top 200 m is 0.20±0.45 ºC and 0.5±0.16 kg $m^{-3}$. Since these effects are small and limited to the surface, we conclude that neither nudging nor the neglect of horizontal
advection affects our conclusions significantly.

### 4.2.2 Effects of $N_2$ Fixation on DIP and DIN

Figure 4 shows simulated $NO_3$ and $PO_4$ concentrations from models H0', H0, H1, H2, H3' and H3, along with the corresponding measurements. Observed $NO_3$ and $PO_4$ concentrations exhibit a marked increase in deep water after the strong winter mixing of 2008. Weaker winter mixing after 2008 results in deep-nutrient accumulation, which is more pronounced for
$NO_3$ than $PO_4$. Model H0' reproduces some deep-nitrate accumulation, but underestimates $NO_3$ concentrations in comparison to observations. Model H0 strongly underestimates inorganic nitrogen below the nutricline. Model H1, where $N_2$ fixation was introduced via a generic autotroph, generates only small changes in the vertical distribution of nutrients. In model H2 the representation of $NO_3$ below the nutricline is slightly improved; however, underestimation of mid-water $NO_3$ is still noticeable. Model H3 significantly improves the representation of deep $NO_3$ accumulation. All model versions represent similar vertical
distributions of $PO_4$ and underestimate its deep-water concentrations by the end of the series.

These model differences are also summarized in Figure 5, which shows the simulated and observed $NO_3$ and $PO_4$ inventories in surface and deep waters. According to the observations deep $NO_3$ accumulates between 2007 and 2010 at a rate of 0.59 ± 0.08 mmol $m^{-2}$ $d^{-1}$, whereas deep $PO_4$ accumulates at 0.015 ± 0.009 mmol $m^{-2}$ $d^{-1}$. During this accumulation period approximately 36 mmol $NO_3$ per mmol $PO_4$ appear in deep waters. All five model versions simulate similar magnitudes and
temporal variability of $PO_4$, but $NO_3$, in particular below 100 m, diverges over time among the models. H0 has the largest deviations from the other models, simulating approximately constant deep $NO_3$ after 2007. H0', the version without

denitrification, produces a rate of increase in deep $NO_3$ similar to that of model version H2. H3 has the highest accumulation rate of deep $NO_3$, matching the observed slope the best.

The $PO_4$ versus $NO_3$ plots in Figure 6 visualize these results in terms of N* values. Observed N* values above 200 m depth can become negative at low nitrate concentrations but are positive at intermediate nutrient concentrations. Below 200 m, observed N* values are positive. This observed pattern in the distribution of nutrients and N* values is not replicated by models H0 and H1. In H0 simulated N* values do not deviate from zero. In H1 N* is consistently positive. As already seen in Figure 4, neither of the two produces large enough nutrient concentrations in deep waters. N* values in H2 qualitatively approach the observed pattern, but maximum nutrient concentrations in deep waters remain too low. Model H3, where heterotrophic diazotrophs co-exist with colonial and unicellular autotrophic diazotrophs, is best able to replicate the range of $NO_3$ and $PO_4$ concentrations and the pattern of N*, although N* values are lower than observed especially at high nutrient concentrations.

Of the additional model versions based on H3 (H3a, H3b, H3c and H3d), results from H3a (heterotrophic and colonial diazotrophs) come closest to H3, while the model without autotrophic diazotrops has the narrowest and most unrealistic range of nutrient concentrations (not shown).

### 4.2.3 Sensitivity to Planktonic Stoichiometry

To investigate whether the vertical distribution of dissolved inorganic nutrients was affected by our assumption of fixed N:P phytoplankton and diazotroph ratios, we explored the sensitivity of $NO_3$ and $PO_4$ to changes in the non-$N_2$-fixing phytoplankton N:P ratio ($R_{N:P}^{nf}$= 16) and the diazotrophs N:P ratio ($R_{N:P}^{f}$= 45). In this analysis we used model version H1, which includes a single non-$N_2$-fixing group and a single generic $N_2$ fixing group and varied the ratios one-at-a-time. As all diazotrophic groups share some characteristics of the phosphate uptake parameterizations, the behaviour of the generic diazotroph is indicative of potential effects in the most extensive model versions in a simplified context. The range of values for $R_{N:P}^{nf}$ varied from 10 to 28 and the range for $R_{N:P}^{f}$ from 19 to 59. Figure 7 shows examples of the results obtained by increasing and decreasing each ratio.

Changes in the N:P ratios had negligible effects on the vertical distribution of $NO_3$, but strongly affected $PO_4$ distribution. In general, lower than Redfield $R_{N:P}^{f}$ of diazotrophs increases $PO_4$ below 300 m, most strongly at depth. This occurs possibly because more phosphorus returns to the dissolved pool per unit nitrogen trough excretion and remineralization. In contrast, a decrease of more than half the $R_{N:P}^{nf}$ of non-diazotrophs produces only a minor decline in deep $PO_4$, while increases in the ratio did not have a significant effect. The decline in deep $PO_4$ occurs because, in the absence of nitrogen limitation, diazotrophs can utilize additional phosphorus.

### 4.2.4 Effect of $N_2$ Fixation on Chlorophyll and $O_2$

Figure 8 shows simulated and observed chlorophyll and dissolved oxygen values. The seasonal variability of total chlorophyll concentrations is reproduced well by all models, with higher chlorophyll between November and April. During these months, simulated chlorophyll concentrations are homogeneous down to 200 m. In 2007 and 2008, chlorophyll concentrations of ~0.13 mg m$^{-3}$ are observed in the measurements reaching as deep as 500 m. This feature is also captured well by our models, as is the location of the deep chlorophyll maximum (DCM) at ~80 m between March and October. However, there are some discrepancies between model results and observations. The models overestimate spring bloom peak concentrations in 2007 and predict peak timing two months earlier than observed in 2008. Model H0 tends to underestimate chlorophyll concentrations from the surface to the DCM during summer months. As chlorophyll concentrations are extremely low during this time of the year, these model-data differences are on the order of 0.05 to 0.1 mg m$^{-3}$. These discrepancies during summer months are corrected in the models with $N_2$ fixation.

Simulated oxygen concentrations exhibit larger differences between models and observations, in particular below the mixed layer, where air-sea fluxes do not directly affect oxygen concentrations. Model versions without diazotrophs (H0 and H0') show similar deep-oxygen variability, with a small underestimation of oxygen during the winter of 2007, and a small overestimation after the winter of 2008. Model re-calibration for the model versions with diazotrophs results in changes in deep oxygen. H1, the model with generic diazotrophs, exhibits the largest model-observation misfits. As in the case of deep $NO_3$, the best deep-oxygen representation is obtained with H3.

### 4.2.5 Validation against Independent Observations

Observations from 2010 to 2014 (outside the optimization period) are used to independently validate the models. The root-mean-square errors (RMSEs) in Table 4 show that, in terms of chlorophyll, $PO_4$ and surface $O_2$, all models behave similarly and achieve similar agreement for assimilated and independent observations. As demonstrated in the previous sections, the model versions mainly diverge in their behaviour with respect to $NO_3$, with some differences in $O_2$ concentrations. Between 0 and 100 m, H3 has the largest RMSEs for $NO_3$, but below 100 m it has the lowest values, particularly against unassimilated $NO_3$. H3 also has the lowest RMSEs for surface and deep oxygen (Table 4).

Figure 9 shows observed and simulated $NO_3$ inventories in 0 – 100 m and below 100 m outside the assimilation period. Compared against the other model versions, H3 increasingly overestimates surface $NO_3$ over time. However, the deep $NO_3$ inventory is best represented by H3. By the end of the observed time series, between 2013 and 2014, H3 starts to also overestimate deep $NO_3$.

### 4.2.6 Primary Production and $N_2$ Fixation Rates

We now compare the simulated rates of primary production with those reported for the Gulf of Aqaba by Rahav et al. (2015) and Iluz et al. (2009) (Figure 10a) and the simulated rates of $N_2$ fixation with those measured by Rahav et al. (2015) and Foster et al., (2009) (Figure 10b). Following Rahav et al. (2015), we show the rates at the DCM and their averages above and below the DCM. The depth-resolved in situ primary production rates reported by Iluz et al. (2009) were also averaged in the same

way for comparison. Where necessary, observations of primary production in carbon units were converted to the model's nitrogen units using the Redfield ratio.

Simulated primary production above the DCM ranges from 0.02 to 0.85 mmol N $m^{-3}$ $d^{-1}$ and exhibits an annual cycle with peaks of productivity in October and April. A prolonged period of low primary production extends from April to September in most model versions. Model versions H3b and H3d maintain rates twice as large as the rest of the models during the summer/fall period. Aside from H3b and H3d, differences between models are small and simulated rates agree with those measured by Iluz et al. (2009) and Rahav et al. (2015).

Above the DCM, models H1, H2, H3 and H3a show a well-defined $N_2$ fixation peak during summer months (i.e., after the peak in primary production). Maximum rates in these models range from 0.001 to 0.1 mmol N $m^{-3}$ $d^{-1}$, which agrees with the observed rates by Foster et al. (2009) and Rahav et al. (2015). In general, simulated $N_2$ fixation rates are low during winter and spring. Similar temporal patterns and differences between model versions occur at the DCM and below. Peaks in $N_2$ fixation at these depth levels occur after the surface peak and have a shorter duration and smaller amplitude. Deep $N_2$ fixation rates estimated by models without heterotrophic diazotrophs do not match the observed rates by Rahav et al. (2015).

## 5 Discussion

### 5.1 Is $N_2$ Fixation Relevant in the Gulf of Aqaba?

In this study we implemented and optimized a series of models with different assumptions about $N_2$ fixation in the Gulf of Aqaba. The models range from one neglecting $N_2$ fixation to another assuming that, in addition to two autotrophic diazotroph groups, $N_2$ fixation can occur in the entire water column (i.e., independent of light availability). While the models are very similar in their abilities to replicate chlorophyll and $PO_4$, model H3 performed the best in reproducing the observed pattern of deep-$NO_3$ accumulation and $O_2$. Overall, all models that consider $N_2$ fixation accumulate nitrogen at different rates, as they enrich the nitrogen content of detritus, which is then remineralized at depth over time.

The best model performance was obtained with two groups of autotrophic organisms and a group of heterotrophic organisms (H3). A model without explicit $N_2$ fixation, but in the absence of sediment denitrification, also increases the accumulation of deep $NO_3$ in a similar fashion as version H2 but not sufficiently high to match the observations. This suggests that $N_2$ fixation in the area must exceed denitrification rates. In the models with denitrification, the average sediment denitrification flux is $0.25 \pm 0.46$ mmol N $m^{-2}$ $d^{-1}$, with a maximum value of 3.01 mmol N $m^{-2}$ $d^{-1}$. These values at are the lower end of a global compilation of sediment denitrification rates by Fennel et al. (2009), which have a mean of 2.2 mmol N $m^{-2}$ $d^{-1}$ and maximum values exceeding 10 mmol N $m^{-2}$ $d^{-1}$.

Observations from the Gulf of Aqaba exhibit an excess of nitrogen that contrasts to exterior waters from the Arabian Sea and Indian Ocean, which are considered net nitrogen sink regions (Gruber and Sarmiento, 1997). As demonstrated by the N* values show in this study, excess nitrogen in the Gulf of Aqaba varies seasonally in surface waters but is more prevalent in deep

waters. We showed that the models' levels of performance at replicating vertical $NO_3$ distributions is directly linked to their ability to reproduce N*. When we neglect $N_2$ fixation, excess phosphate tends to dominate the whole water column, contrary to the observations. Explicitly accounting for $N_2$ fixation (H1, H2, H3) improves models' abilities to replicate N*. Nevertheless, the model's limitations at replicating $PO_4$ also affect their ability to fully simulate the range of observed N*

values. This suggest that the model requires further improvements in the representation of processes differentially affecting $PO_4$ (see section 5.4).

There are too few reported values of dissolved inorganic nitrogen and phosphorus for the Red Sea region from Bab-el-Mandeb to the Strait of Tiran to provide a complete idea of the spatial distribution of N*; however, the limited available information supports our conclusions. Naqvi et al. (1986) found a significant difference in N:P ratios between surface incoming and sub-

surface outflowing waters at Bab-el-Mandeb, concluding that $N_2$ fixation was a process required to explain these anomalies in the nitrogen budget. Higher nitrogen concentrations (N* = +2.5 mmol m$^{-3}$) have been observed in the Red Sea in comparison to the Arabian Sea and Indian Ocean, where a strong deficit of nitrogen develops as losses due to denitrification exceed the input of newly fixed nitrogen (Burkill et al., 1993; Naqvi, 1994; Gruber and Sarmiento, 1997; Morrison et al., 1998, 1999). Close to the entrance of the Persian Gulf, average N* values have been estimated to be below -5 mmol m$^{-3}$ at all depths and

seasons reported, with minimum N* values on the order of -8 mmol m$^{-3}$ (Gruber and Sarmiento, 1997). Thus, it has been hypothesized that limited deep-water exchange at Bab-el-Mandeb allows waters of the Red Sea outside of the Gulf of Aqaba to acquire different characteristics from inflowing Arabian Sea waters (Naqvi et al., 1986). Our model results support this hypothesis and suggest that $N_2$ fixation is key for the formation of the distinct biochemical characteristics in the Gulf of Aqaba. Considering the regional context, our models suggest that, despite low rates, $N_2$ fixation is necessary to explain the nitrogen

vertical distribution in the Gulf of Aqaba, and the interannual accumulation of deep nitrate during years with weak convection.

The lowest negative N* values observed in surface waters in the Gulf of Aqaba during summer are not fully captured by any of our model versions. In the context of a one-dimensional framework, we cannot reject the possibility that these minimum N* values are a remnant signal of denitrification in the distant Arabian Sea. During their passage through the Red Sea, $N_2$ fixation may be responsible of transforming waters with significant excess phosphorus into these summer surface waters with small

negative N* deviations. If we consider the global average correction to N* values of +2.89 µmol kg$^{-1}$ used by Gruber and Sarmiento (1997), N* values in the Gulf of Aqaba hold a permanent excess of nitrate with respect to other geographical regions. Similarly, the overestimation of surface $NO_3$ obtained with the model that performs the best for deep $NO_3$ suggests that the Gulf has potential to export newly fixated nitrogen to the outside waters through horizontal advection in the surface to mid-water layers. This was not tested within our one-dimensional model. Given our model results in the context of these regional

characteristics, we consider that $N_2$ fixation is a necessary to explain positive N* values in the Gulf of Aqaba, and the interannual accumulation of deep nitrate during years with weak convection.

### 5.2 How does $N_2$ fixation Contribute to Primary Production?

In this section we discuss the contribution of $N_2$ fixation to primary production in the Gulf of Aqaba, and our quantitative estimates of $N_2$ fixation with respect to global rates (Figures 9-11). Our estimates of surface primary productivity agree with those reported by Iluz et al. (2009) for March-April of 2008. However, our models overestimate surface primary productivity values in 2010 when compared to those reported by Rahav et al. (2015). On average, our best-performing model version yields

an annual primary production rate of $304\pm56.9$ g C m$^{-2}$ yr$^{-1}$ (H3). This rate is higher than previously published annual averages, which range from 80 g C m$^{-2}$ y$^{-1}$ (Levanon-Spanier et al., 1979) to 170 g C m$^{-2}$ y$^{-1}$ (Iluz, 1991), whereas more recent unpublished primary production estimates at IUI range between 141 and 197 g C m$^{-2}$ y$^{-1}$ (pers. comm. Y. Shaked).

The ratio of new to total primary production (f-ratio) in our model experiments ranges from 15% to 80%. Maximum f-ratios are estimated in January and February due to significant contributions from deep $NO_3$, whereas f-ratios are at their minimum

during stratified conditions (June – August). Our best-performing model version, H3, estimates a summer minimum f-ratio 0.22. The average f-ratio for all scenarios is 0.47. This agrees with published estimates for the Gulf of Aqaba of 0.5 during the stratified period as determined from a nitrate-diffusion model (Badran et al., 2005).

Total annual $N_2$ fixation rates from our best-performing model versions (H3 and H3a) are similar to high estimates reported for other regions (Capone and Carpenter, 1982; Michaels et al., 1996; Lee et al., 2002), while those obtained in the other model

experiments are within the range of values reported for the Gulf of Aqaba. The intensity of winter mixing has a minor effect on $N_2$ fixation rates; the largest effect occurred in H3a where $N_2$ fixation increased by 15% after deep winter mixing. Based on our best-performing model version (H3), we estimate that 10% to 14% of the total primary production is supported by $N_2$ fixation.

### 5.3 Are Heterotrophic $N_2$ Fixers Important?

In contrast to previous models (e.g., Hood et al., 2001; Fennel et al., 2002; Monteiro et al., 2010; Moore et al., 2004), our model version H3 relaxes the assumption of light dependence for diazotrophy, through the inclusion of heterotrophic diazotrophs in addition to two groups of autotrophic diazotrophs. This model improves the representation of $NO_3$ and $O_2$ at depth (Figures 3, 6). Changes in deep $NO_3$ can be explained through the enrichment of detritus, while changes in $O_2$ may reflect the additional sink of $O_2$ at depth due to the heterotrophic group. All model versions with heterotrophic organisms also

match observed estimates of $N_2$ fixation in deep waters of the Gulf of Aqaba, and without them $N_2$ fixation rates below the DCM are underestimated (Figure 8). Heterotrophic $N_2$ fixation also impacts total $N_2$ fixation (Figure 9).

There is growing evidence of non-cyanobacterial $N_2$ fixation in aphotic waters (Benavides et al., 2017; Moisander et al., 2017). For instance, $N_2$ fixation rates in mesopelagic and abyssopelagic waters down to 2000 m (Fernandez et al., 2011; Bonnet et al., 2013; Loescher et al., 2014) have been attributed to non-cyanobacterial organisms, including proteobacteria (Turk-Kubo et al.,

2014). These *nif*H-expressing heterotrophic phylotypes can be as abundant as unicellular cyanobacterial groups and dominate the deep and dark zones of the water column (Church et al. 2005; Langlois et al., 2005; Riemann et al., 2010). Genetic evidence and rate estimates from the Gulf of Aqaba suggest that *nif*H expressing heterotrophic proteobacteria α and γ may explain the

correlation of bacterial productivity rates with $N_2$ fixation rates (Rahav et al., 2013; 2015). Aside from the Gulf of Aqaba, aphotic $N_2$ fixation and *nif*H gene expression have also been reported in the Baltic Sea (Farnelid et al., 2013), Arabian Sea (Jayakumar et al., 2012) and Mediterranean Sea (Rahav et al., 2013).

Despite the importance of heterotrophic diazotrophs in our model, the simulated colonial diazotroph blooms are responsible for the highest $N_2$ fixation rates, so they are a necessary model aspect to achieve resemblance with the observations. This is in line with evidence of extensive blooms of *Trichodesmium spp.* being responsible for the high $N_2$ fixation rates observed in the Arabian Sea and Red Sea (Capone et al., 1998; Post et al., 2002; Foster et al., 2009). In the northern Gulf of Aqaba, colonies and free trichomes of *Trichodesmium spp.* are found throughout the year down to 100 m depth (Post et al., 2002). Ephemeral blooms of *T. erythraeum* and *T. thiebautii* have been documented near the coast of Eilat (Post et al., 2002, Gordon et al., 1994; Kimor and Golandsky, 1977). However, massive blooms are rare in the Gulf of Aqaba (Foster et al., 2009; Mackey et al., 2007, pers. Communication Berman-Frank) and the model probably overestimates the contribution of *Trichodesmium spp.*'s annual blooming to total $N_2$ fixation rates, as seen in the much larger surface $N_2$ fixation rates generated by H2 and H3. As new observational information is collected, further model refinements may be necessary to better reflect the actual contribution of different diazotrophic groups in the Gulf of Aqaba.

## 5.4 Limitations and Uncertainties

The one-dimensional nature of our physical setting, which neglects the contribution of horizontal advection to the vertical structure of simulated tracers, can be considered a limitation of this study. This simplification is, however, necessary to perform model calibration and test multiple model structures at a manageable computational expense. We applied temperature and salinity nudging to ensure accurate representation of the vertical density structure. Comparison of the simulated vertical structure with and without nudging shows that this correction has negligible effects on deep waters, where the effects of $N_2$ fixation are the most relevant. This is consistent with the existing literature about circulation of the Gulf of Aqaba, which describes how geomorphology and bathymetry limit water exchange between the Gulf of Aqaba and the Red Sea to the upper 300 m (Wolf-Vecht et al., 1992; Biton and Gildor, 2011). It is, therefore, unlikely that horizontal transport could explain the observed accumulation of deep $NO_3$. Nevertheless, transport of nitrogen-enriched sub-surface waters from the Gulf of Aqaba towards the exterior may dampen and modulate the long-term accumulation of nitrogen observed in Figure 9.

There are other sources of nitrogen that were not explored in the present study that we discuss here briefly. For instance, we did not include contributions to $N_2$ fixation by diatom-diazotroph associations, which are significant in other regions. While diatom-diazotroph associations have been detected in the Gulf of Aqaba, they are not as abundant as unicellular diazotrophs, *Trichodesmium* and proteobacteria (Kimor et al. 1992, Foster et al., 2009, pers. Comm Berman-Frank). In general, due to the oligotrophic characteristics of the region, small phytoplankton species (<8 μm) contribute more than 90% of the chlorophyll-a standing stock (Lindell & Post 1995, Yahel et al. 1998). Dinoflagellates and diatoms together correspond to less than 5% of the phytoplankton biomass, except during ephemeral diatom blooms during spring when they can account for nearly 50% of

the total biomass (Al-Najjar et al., 2006).

Another source of nitrogen which could significantly affect this region is atmospheric deposition, as the Gulf receives considerable dust input from the surrounding deserts. Recently, it has been shown that atmospheric dust input does not correlate with chlorophyll variability in surface waters of the Gulf of Aqaba (Torfstein and Kienast, 2018). In contrast, a previous study
suggested that atmospheric deposition of nitrogen could support over 10% of surface primary production in the region, based on measurements of local aerosol composition and a dust deposition model (Chen et al., 2007). However, this estimate had a relatively large uncertainty due to errors associated with the deposition flux calculation and the temporal variability in dust flux (Chen et al., 2007). Moreover, very low nitrogen concentrations and N:P ratios lower than Redfield from the surface down to 80 m were observed during the same time period (Foster et al., 2009). Therefore, the role of atmospheric nitrogen inputs
remains uncertain.

In contrast to the improvement in $NO_3$ distributions with the addition of heterotrophic diazotrophs, all other model versions exhibit similar underestimation of deep total $PO_4$. This suggests that their structure lacks a process affecting this nutrient. Several processes can independently affect $PO_4$. Prior knowledge suggests that seasonal advection of nitrogen depleted surface waters may contribute to changes in $PO_4$, while having minimum impact on $NO_3$ concentrations. Other relevant missing
processes include variable plankton stoichiometry and $PO_4$ remineralization. Our sensitivity analysis highlighted that varying the planktonic N:P ratios has an evident effect on $PO_4$ but does not affect $NO_3$ distributions significantly. A related effect was noticed when comparing fixed constant vs. variable plankton nutrient uptake stoichiometry in a model of the central Baltic Sea: seasonal particulate organic C:N ratios were very similar, but C:P diverged (Kreuss et al., 2015). Moreover, preferential $PO_4$ remineralization may more directly increase deep-$PO_4$ concentrations as organic matter is decomposed in the bottom
layers. The inclusion of this process in a model of the North Atlantic Ocean improved the representations of biogeochemical characteristics of the area and increased the $N_2$ fixation rates obtained by the model (Monteiro and Follows, 2012). Variable nitrogen allocation within diazotrophs also improved the performance of a model representing annual cycles in the North Atlantic sub-tropical gyre (Fernández-Castro et al., 2016). Further investigation of this process is recommended.

Finally, an intrinsic caveat that should accompany all biological models is the uncertainty associated with parameter values
(Denman, 2003). We reduced this uncertainty by using systematic parameter optimization for a subset of parameters, yet others were kept fixed at a priori assumed values.  In the case of N:P ratios, a simple sensitivity analysis suggests that they do not affect conclusions with respect to inorganic dissolved nitrogen but modify phosphorus concentrations. In the case of parameters related to diazotrophs, these are largely unconstrained by the observations we used, which are the typically available ones in long-term data sets (i.e., Chl-a, $NO_3$, $PO_4$, $O_2$). We emphasize that results in this study are exploratory, testing the effects of
assumptions about diazotroph behaviour rather than modifying such behaviour by subjectively tuning parameter values.

**6 Conclusions**

We implemented and optimized biogeochemical models that represent a range of different assumptions about diazotrophy in a 700 m-deep pelagic station from the northern Gulf of Aqaba. Our model results demonstrate the importance of $N_2$ fixation in replicating the observed water-column-integrated nitrogen and oxygen inventories. The model without $N_2$ fixation is unable to replicate the observed vertical structure of inorganic nitrogen. The models that include diazotrophs significantly modify this variable by increasing the fraction of remineralized nitrogen from organic matter decomposition. The effect of $N_2$ fixation on $O_2$ distributions depends on the type of nitrogen fixer. $N_2$ fixation by autotrophic-photosynthetic organisms increases oxygen concentrations, while heterotrophic organisms decrease deep-oxygen due to increased respiration and organic matter decomposition. The observed vertical structure of $NO_3$ and oxygen is reproduced best with a model that includes heterotrophic, and colonial and unicellular autotropic diazotrophs suggesting that heterotrophic $N_2$ fixation is necessary to explain the observed excess nitrogen at this location. $N_2$ fixation assumptions do not affect $PO_4$ concentrations significantly, but they are affected by assumptions about N:P ratios of organic matter. The $N_2$ fixation rates simulated by this model are similar to the highest observational estimates from the Gulf of Aqaba. Aphotic $N_2$ fixation is simulated to occur at lower rates than maximum autotrophic $N_2$ fixation yet occurs continuously over a large portion of the water column. This suggests that heterotrophic diazotrophs set a background rate of $N_2$ fixation in the ocean that should be considered further in global estimates and biogeochemical models.

**Acknowledgements** We thank the Inter University Institute for Marine Sciences in Eilat and A. Genin for providing the observational data, and E. Landou, B. Lazar (BL), S. Kienast, H. Gildor, E. Boss, A. Torfstein, and M. Lewis (ML) for insightful discussions. We gratefully acknowledge financial support from the Schulich Marine Studies Initiative to KF, IBF, ML, and BL.

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

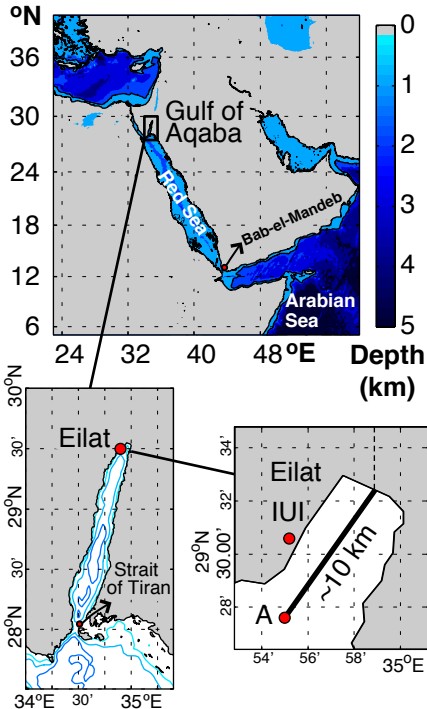

**Figure 1: Map of study area showing the location of monitoring stations and geographic references.**

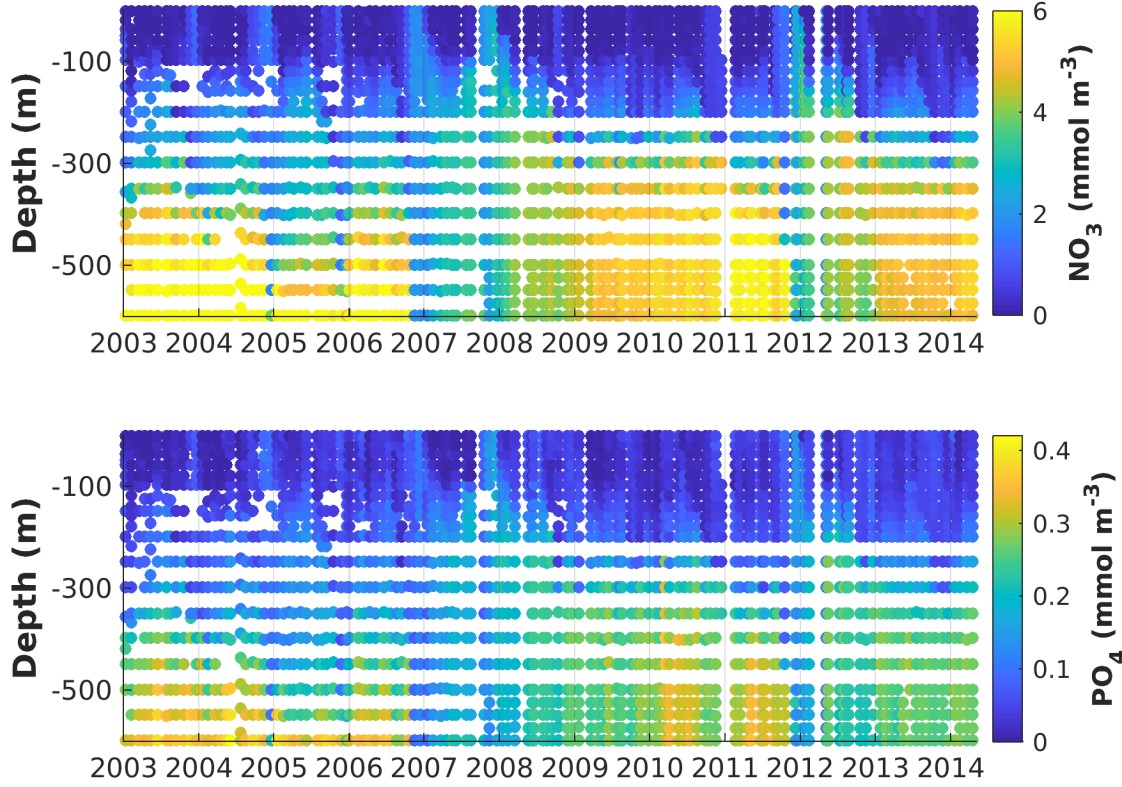

**Figure 2: Observed vertical nitrate and phosphate distributions at Station A (Gulf of Aqaba) from 2004 to 2014. Tick marks are placed on the April 1$^{st}$ of every year.**

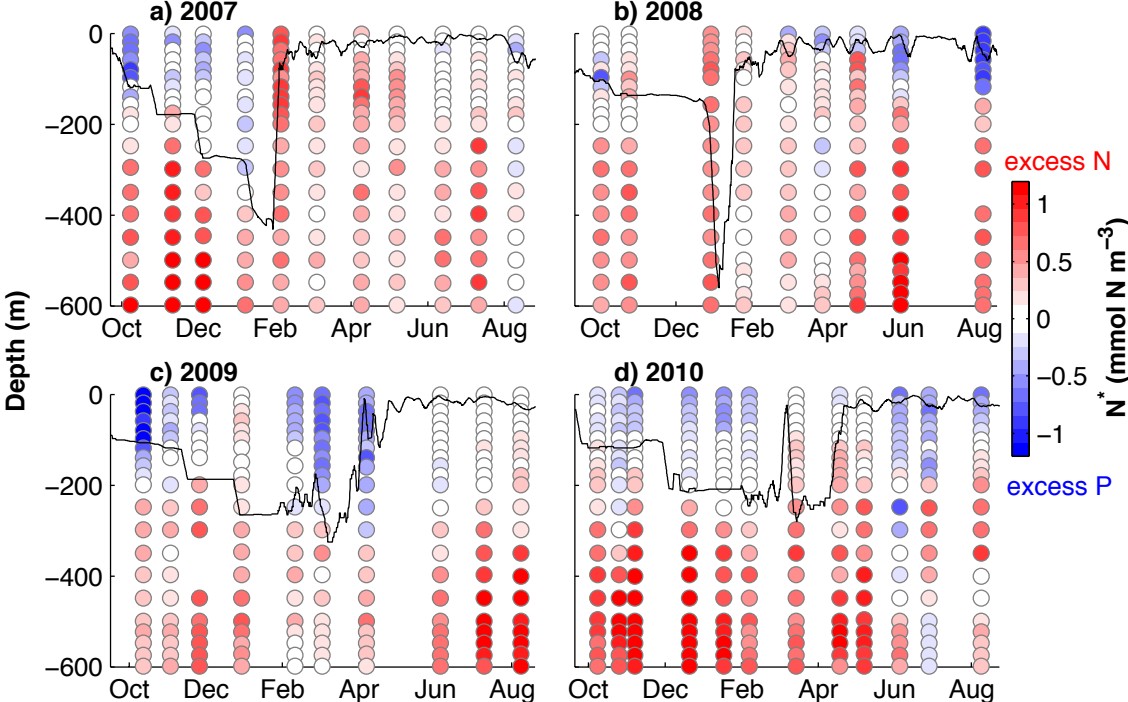

**Figure 3: N\* calculated from NO₃ and PO₄ profiles from Figure 2. The black lines show estimated mixed layer depths using a maximum density gradient criterion.**

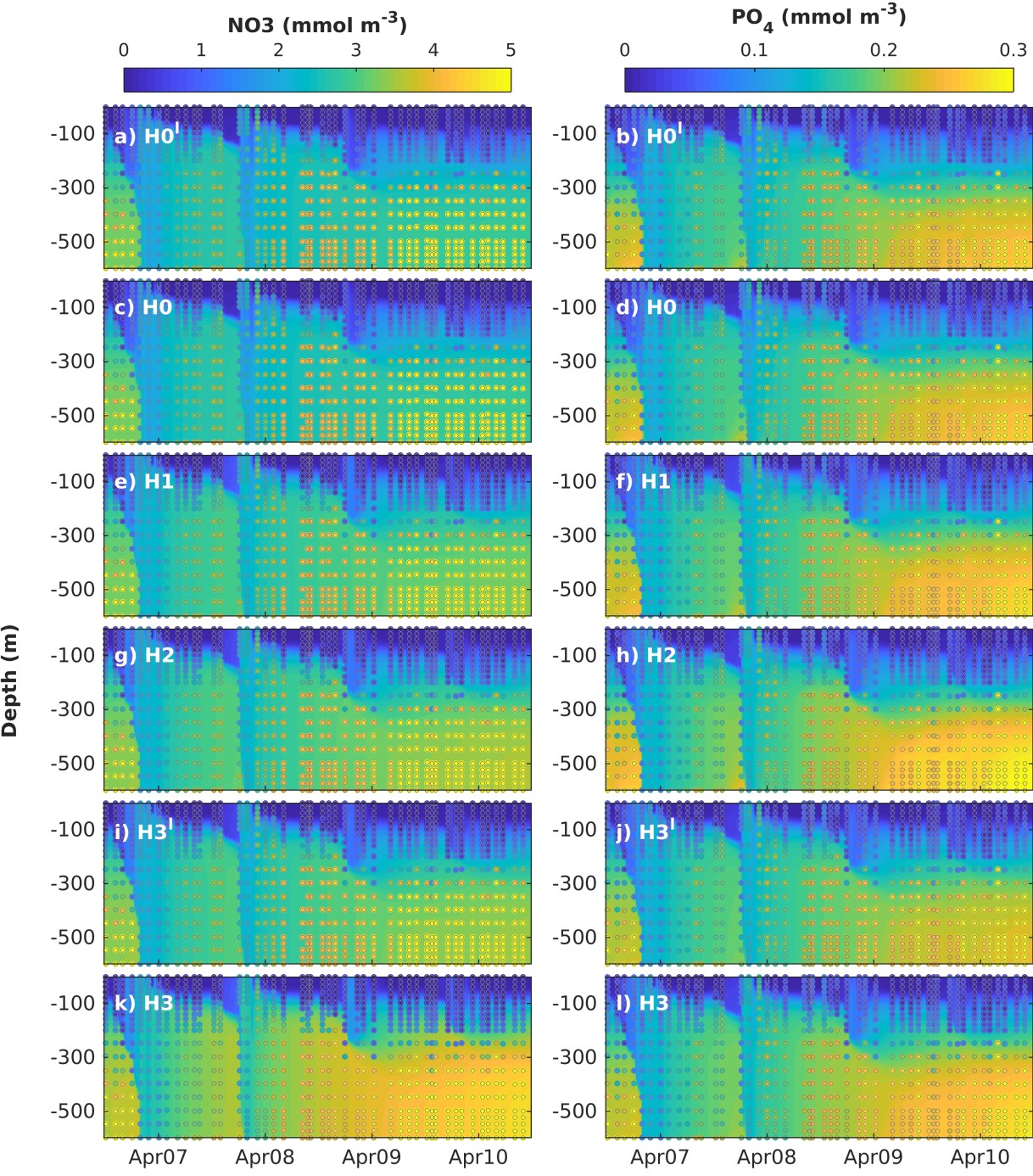

**Figure 4: Observed (coloured circles) and simulated (background) NO₃ and PO₄ using model versions H0 (no nitrogen fixers), H1 (generic autotrophic fixer), H2 (unicellular and colonial autotrophic fixers), H3' (non-N₂-fixing heterotrophic, and unicellular and colonial autotrophic fixers), H3 (heterotrophic, and unicellular and colonial autotrophic fixers). The spin-up period is not shown.**

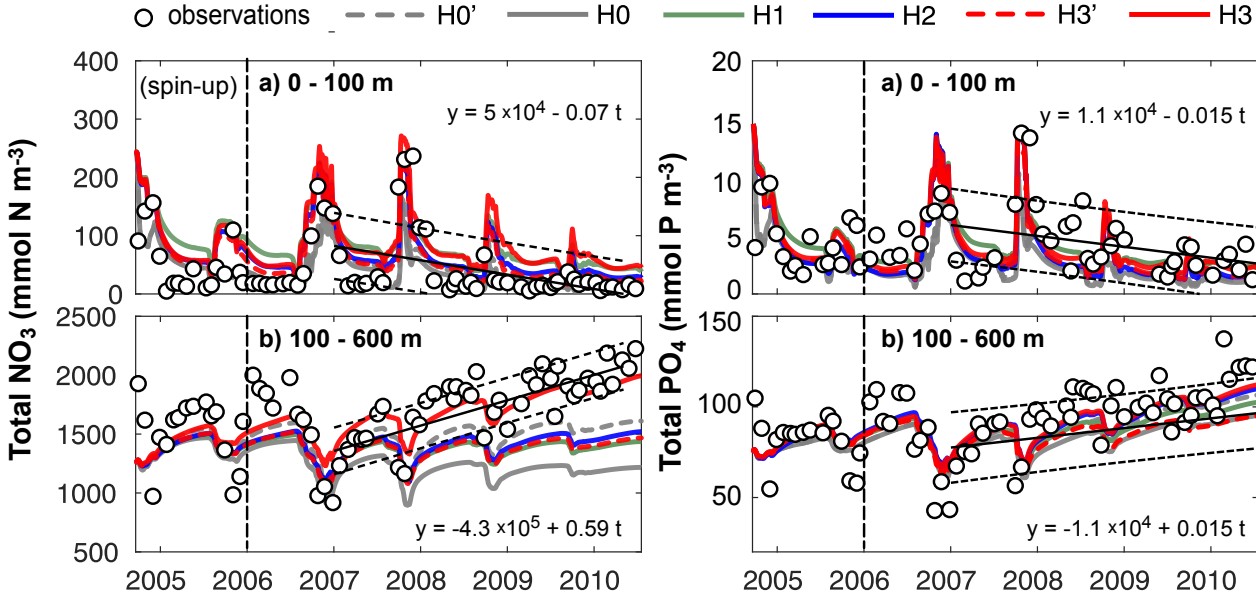

**Figure 5: Observed and simulated vertically integrated NO$_3$ and PO$_4$ between 0 – 100 m and between 100 – 600 m using model versions H0 (no nitrogen fixers), H0' (no sediment denitrification – no fixers), H1 (generic autotrophic fixer), H2 (unicellular and colonial autotrophic fixers), H3' (non-N$_2$-fixing heterotrophic, and unicellular and colonial autotrophic fixers), H3 (heterotrophic, and unicellular and colonial autotrophic fixers). Tick marks are placed on the April 1$^{st}$ of every year.**

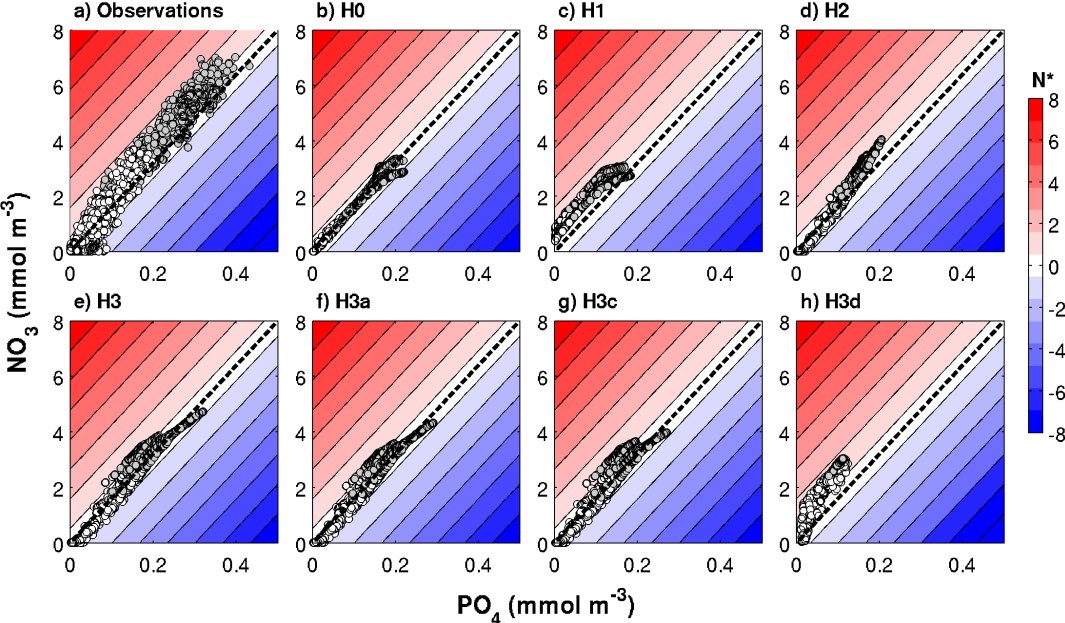

**Figure 6: Observed and simulated N\* range in a subset of the optimized biogeochemical models. The dashed black diagonal line marks the N\* = 0, or N:P = 16 line.**

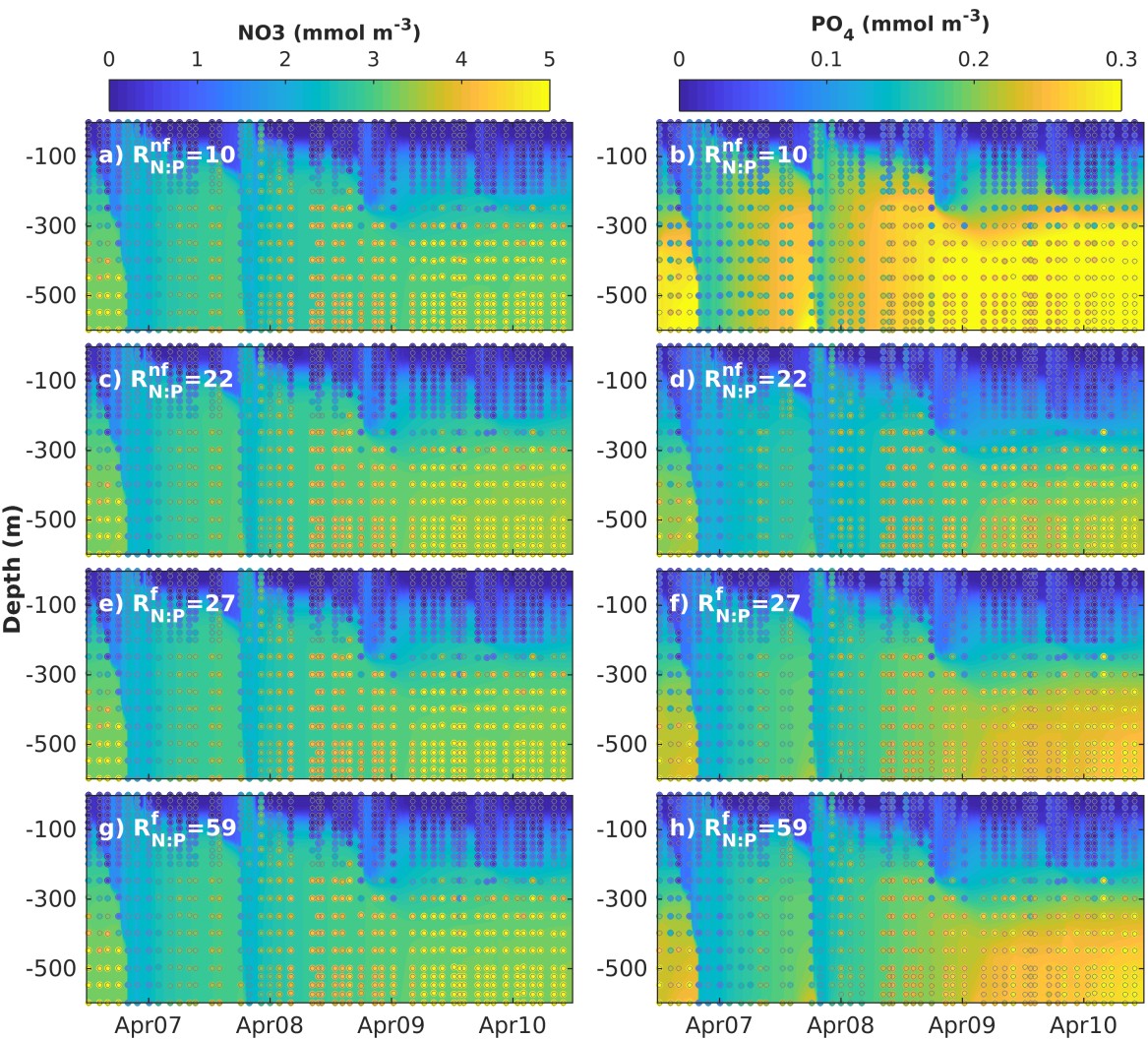

**Figure 7: Results of sensitivity analyses modifying the fixed N:P ratios in non-N₂-fixing phytoplankton ($R_{N:P}^{nf}$) and N₂ fixing generic diazotrophs ($R_{N:P}^{f}$). Observed (coloured circles) and simulated (background) NO₃ and PO₄ using model versions H1 (generic autotrophic fixer). The spin-up period is not shown.**

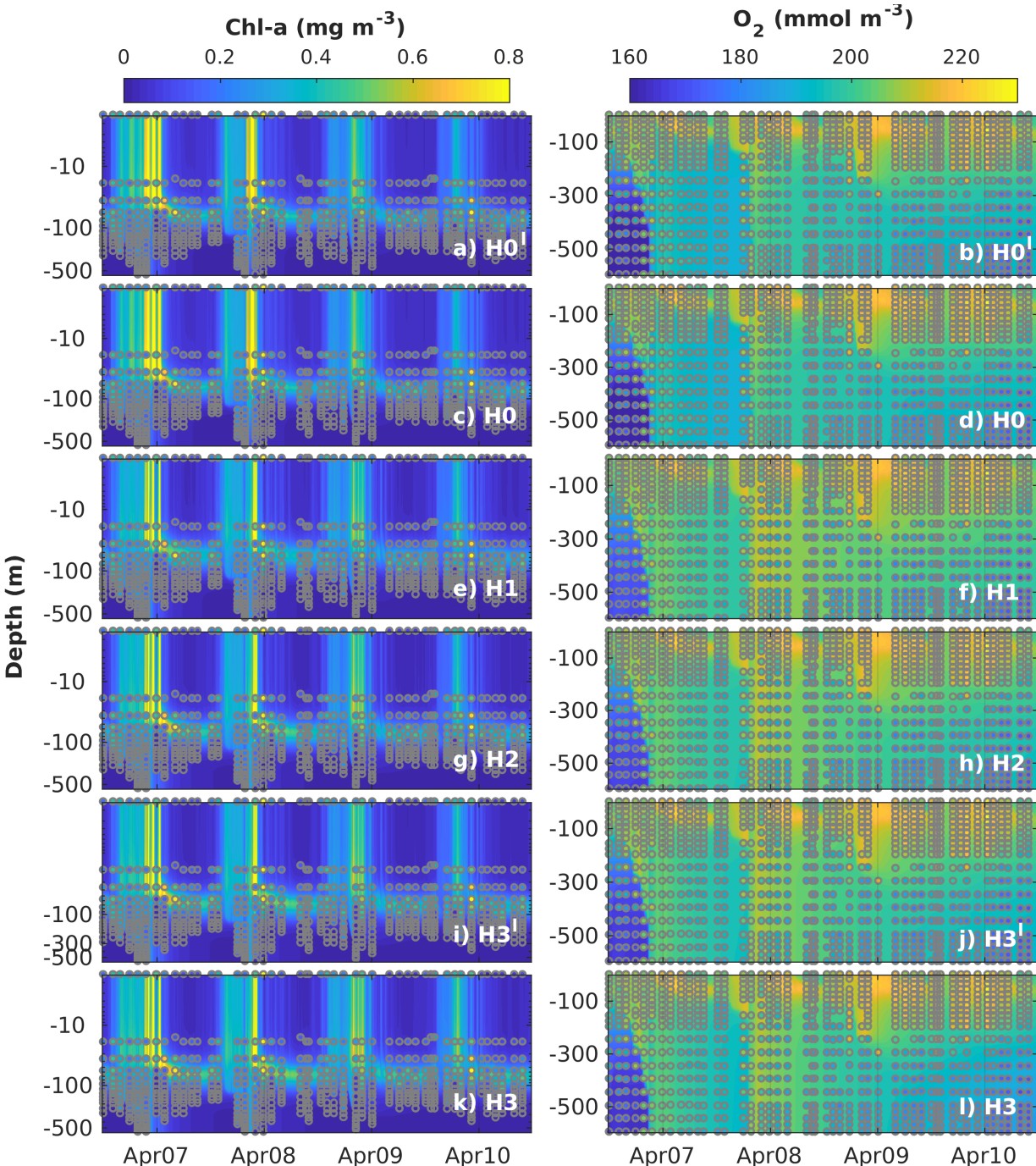

**Figure 8: Observed (coloured circles) and simulated (background) Chl-a and O₂ using model versions H0' (no sediment denitrification – no fixers), H0 (no nitrogen fixers), H1 (generic autotrophic fixer), H2 (unicellular and colonial autotrophic fixers), H3' (unicellular and colonial autotrophic fixers and heterotrophic non-fixer), H3 (non-N₂-fixing heterotrophic, and unicellular and colonial autotrophic fixers), and H3 (heterotrophic, and unicellular and colonial autotrophic fixers). Vertical scale in the Chl-a subplots is logarithmic to exaggerate the surface. The spin-up period is not shown.**

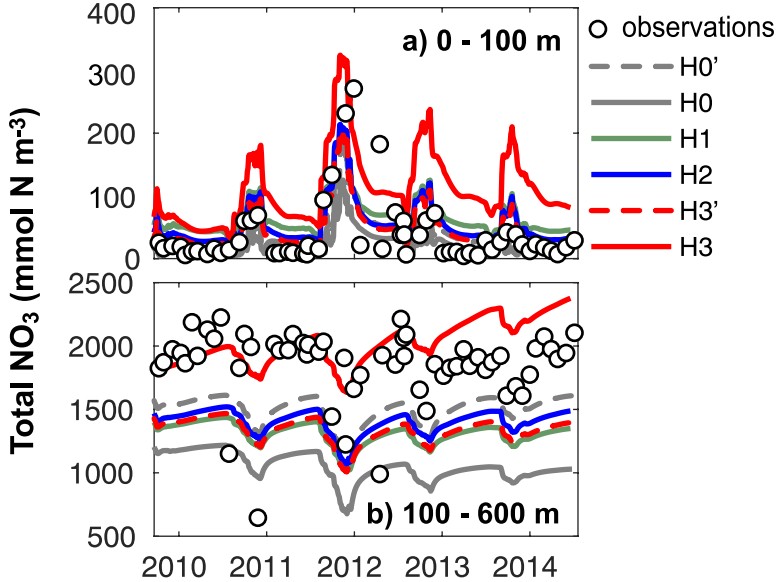

**Figure 9: Observed (circles) and simulated (lines) total nitrate in the surface and deep waters at Station A during the model validation period from 2010 to 2014. Tick marks are placed on the April 1st of every year.**

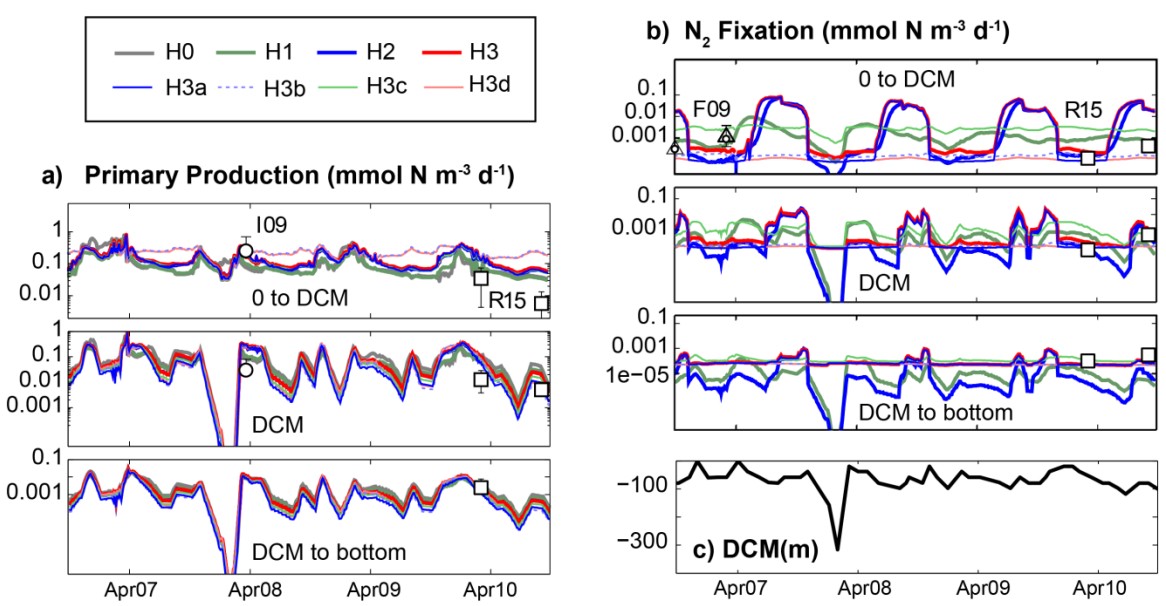

**Figure 10: Comparison of previously reported in situ measurements and model results of primary production (a) and N₂ fixation rates (b), averaged at three depth levels. Depth levels are from the surface to the Deep Chlorophyll Maximum, at the DCM and below it. (c) DCM estimated from observed Chl-a profiles at Station A. I09, F09 and R15 refer to Iluz et al., (2009), Foster et al., (2009), and Rahav et al., (2015), respectively. Control versions H0' and H3' are not shown.**

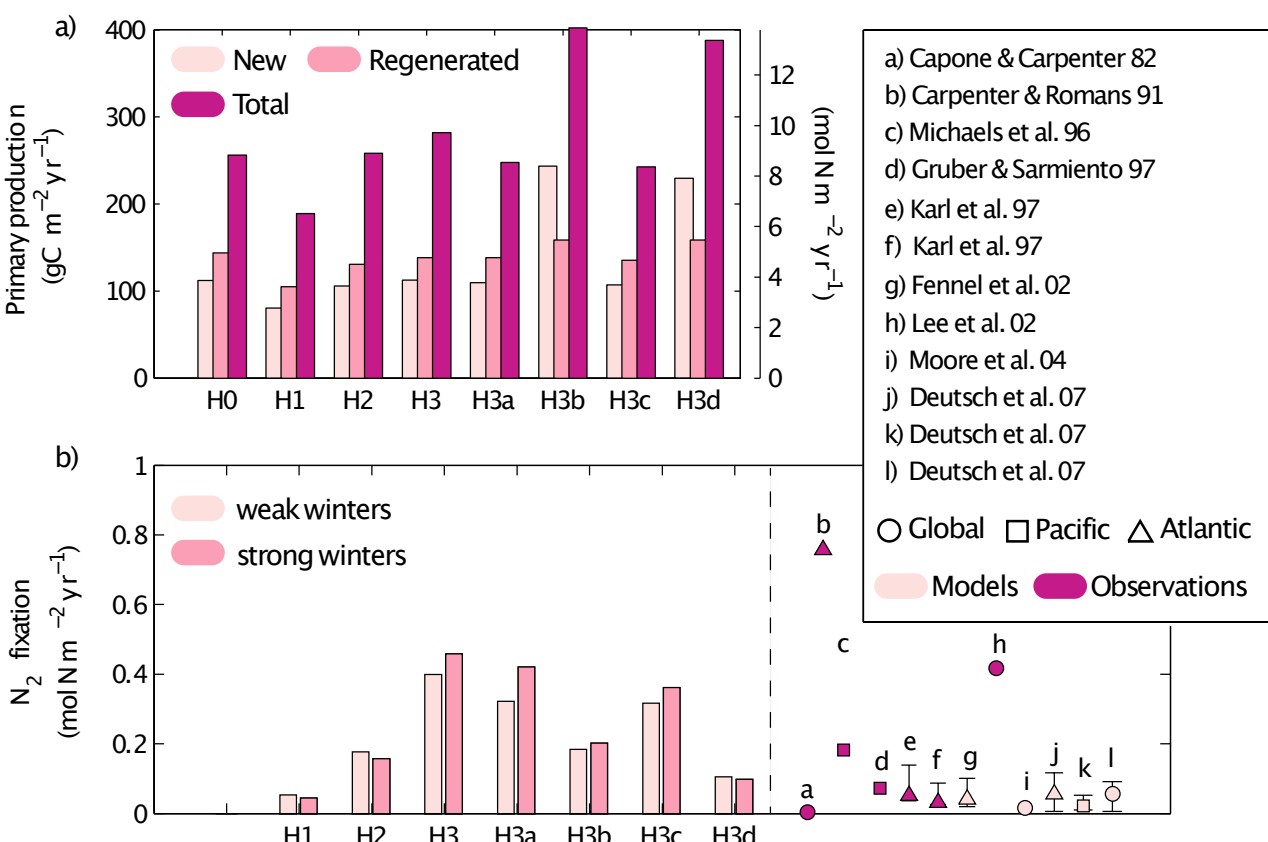

**Figure 11: Simulated new, regenerated, and total primary production (a) and N₂ fixation rates (b) obtained by the different simulations (see Table 1 for key to different simulations). A summary of previous estimates of N₂ fixation rates in observational and model studies is included in (b). Redfield C:N ratio is used to transform the model results nitrogen units to carbon units. Control versions H0' and H3' are not shown.**

**Table 1: Summary of model version characteristics and assumptions about diazotroph groups. N = no diazotrophs; A = autotrophic diazotrophs; H = heterotrophic diazotrophs. Checkmarks (✓) represent presence of a model characteristic / functional diazotrophic group in the model; dashes represent their absence.**

| Model Version | Characteristics / Diazotrophs Groups | Denitrification | Generic (A) | Unicellular (A) | Colonial (A) | Heterotrophic (H) |
|---|---|---|---|---|---|---|
| **H0'** | N | - | - | - | - | - |
| **H0** | N | ✓ | - | - | - | - |
| **H1** | A | ✓ | ✓ | - | - | - |
| **H2** | A | ✓ | - | ✓ | ✓ | - |
| **H3'** | A H | ✓ | | ✓ | ✓ | Non-N$_2$-fixing |
| **H3** | A H | ✓ | - | ✓ | ✓ | ✓ |
| **H3a** | A H | ✓ | - | - | ✓ | ✓ |
| **H3b** | A H | ✓ | - | ✓ | - | ✓ |
| **H3c** | A H | ✓ | ✓ | - | - | ✓ |
| **H3d** | H | ✓ | - | - | - | ✓ |
| | **Diazotrophs characteristics:** | | | | | |
| | Inorganic phosphorus uptake | | ✓ | ✓ | ✓ | ✓ |
| | Organic phosphorus uptake | | - | - | - | ✓ |
| | Light growth limitation | | ✓ | ✓ | ✓ | - |
| | Temperature dependent maximum growth rate | | ✓ | ✓ | ✓ | ✓ |
| | Minimum temperature limit for growth (20ºC) | | - | - | ✓ | - |
| | Predation | | - | ✓ | - | - |

**Table 2: Parameters used in the base biogeochemical model (H0), including minimum and maximum parameters ranges based on the literature. Parameters values followed by * were obtained by the optimization.**

| Parameters | Value | Range | Description | Units | References |
|---|---|---|---|---|---|
| $\mu_{Phy}^{0}$ | 0.76* | 0.1 – 3 | Reference phytoplankton maximum growth rate at T = 0ºC | $d^{-1}$ | a, b, c |
| $k_{Phy}^{NO_3}$ | 0.05 | 0.01 – 0.5 | Phytoplankton $NO_3$ uptake half-saturation | $mmol\ m^{-3}$ | d, e |
| $k_{Phy}^{NH_4}$ | 0.1* | 0.01 – 0.5 | Phytoplankton $NH_4$ uptake half-saturation | $mmol\ m^{-3}$ | d, e |
| $k_{Phy}^{DIP}$ | 0.004* | 0.001 – 0.5 | Phytoplankton DIP uptake half-saturation | $mmol\ m^{-3}$ | a, f, g |
| $\alpha_{Phy}$ | 0.1* | 0.01 – 0.125 | Phytoplankton, initial slope of photosynthetic response | $molC\ gChl^{-1}\ (W\ m^{-2})^{-1}\ d^{-1}$ | d, h |
| $m_{Phy}$ | 0.1 | 0.01 – 0.2 | Phytoplankton mortality rate | $d^{-1}$ | d |
| $g_{Phy}^{max}$ | 1.16* | 0.1 - 4 | Zooplankton maximum grazing rate | $d^{-1}$ | b, i |
| $k_{Zoo}^{Phy}$ | 0.5* | 0.01 – 0.5 | Square zooplankton grazing half-saturation | $(mmol\ m^{-3})^2$ | d, e |
| $l_{BM}$ | 0.011* | 0.01 – 0.15 | Zooplankton base metabolic rate | $d^{-1}$ | d |
| $l_E$ | 0.1 | 0.05 – 0.35 | Zooplankton excretion rate | $d^{-1}$ | d |
| $m_Z$ | 0.35* | 0.02 - 0.35 | Zooplankton mortality rate | $d^{-1}$ | d |
| $\tau$ | 0.1 | 0.01 - 25 | Small detritus aggregation rate | $d^{-1}$ | d, e |
| $\theta_{Phy}^{max}$ | 0.142* | 0.015 – 0.15 | Maximum chlorophyll to carbon ratio | $mg\ Chl\ (mg\ C)^{-1}$ | h |
| $\beta$ | 0.74* | 0.25 – 0.75 | Zooplankton assimilation efficiency | non-dim. | j, k |
| $r_{DOM}$ | 0.2 | 0.05 – 0.5 | DOM remineralization rate | $d^{-1}$ | l |

| | | | | | |
|---|---|---|---|---|---|
| $r_D$ | 0.01 | 0.005 – 0.15 | Detritus remineralization rate | d$^{-1}$ | l, m |
| $n_{max}$ | 0.3* | 0.01 – 0.35 | Nitrification rate | d$^{-1}$ | d, e |
| $k_I$ | 0.1 | 0.01 – 0.5 | Half-saturation radiation for nitrification inhibition | Wm$^{-2}$ | d |
| $I_{th}$ | 0.0095 | 0.005 – 0.01 | Radiation threshold for nitrification inhibition | Wm$^{-2}$ | d |
| $w_{Phy}$ | 0.1 | 0.01 – 1 | Vertical sinking velocity for non-N$_2$-fixing phytoplankton | md$^{-1}$ | n |
| $w_{DL}$ | -4.44* | 0.01 – 25 | Vertical sinking velocity for large detritus | md$^{-1}$ | d |

a. Fennel et al. (2002) b. Fahnenstiel et al. (1995) c. Veldhuis et al. (2005) d. Fennel et al. (2006) e. (Lima and Doney (2004) f. Ward et al. (2013) g. Moore, et al.(2002) h. Geider et al. (1997) i. Gifford et al. (1995) j. Landry et al. (1984) k. Tande and Slagstad (1985) l. Amon and Benner (1996) m. Enríquez et al. (1993) n. Smayda and Bienfang (1983)

**Table 3: Diazotrophs parameters and re-calibrated non-N₂-fixing phytoplankton parameters for each model version. H0 = no N2 fixers; H1 = generic autotrophic diazotrophs; H2 = unicellular and colonial cyanobacteria; H3 = heterotrophs, unicellular and colonial cyanobacteria. Superscripts in the parameter descriptions denote the corresponding literature references used to define the generic diazotroph parameter values within a realistic range. In order to tease apart the effect of different diazotrophs niches or behaviour, unicellular, colonial and heterotrophic diazotrophs maintain the same parameter values as the generic organism. An exception is the slightly lower reference growth rate used for heterotrophic diazotrophs.**

| Model version: | H0 | H1 | H2 | H3 | Units | Description |
|---|---|---|---|---|---|---|
| $\mu_{Phy}^{0}$ | 0.76 | 2.20 | 1.5 | 1.5 | d$^{-1}$ | Reference phytoplankton maximum growth rate at T = 0ºC |
| $\theta_{Phy}^{max}$ | 0.022 | 0.076 | 0.076 | 0.05 | mg Chl (mg C)$^{-1}$ | Maximum chlorophyll to carbon ratio – non-N₂-fixing phytoplankton |
| $k_{Phy}^{NH_4}$ | 0.076 | 0.076 | 0.076 | 0.076 | mmol m$^{-3}$ | Phytoplankton NH₄ uptake half-saturation |
| $k_{Phy}^{DIP}$ | 0.001 | 0.015 | 0.015 | 0.015 | mmol m$^{-3}$ | Phytoplankton DIP uptake half-saturation |
| $m_{Phy}$ | 0.1 | 0.06 | 0.06 | 0.06 | d$^{-1}$ | Phytoplankton mortality rate |
| $g_{Phy}^{max}$ | 1.16 | 4.0 | 1.95 | 1.95 | d$^{-1}$ | Zooplankton maximum grazing rate |
| $\beta$ | 0.36 | 0.7 | 0.7 | 0.7 | non-dim. | Zooplankton assimilation efficiency |
| $\mu_{G_F}^{0}$ | - | 0.25 | - | - | d$^{-1}$ | Reference generic diazotrophs maximum growth rate at T = 0ºC[a -j] |
| $k_{G_F}^{DIP}$ | - | 0.001 | - | - | mmol m$^{-3}$ | Generic diazotrophs DIP uptake half-saturation[d] |
| $\theta_{F}^{max}$ | - | 0.053 | - | - | mg Chl (mg C)$^{-1}$ | Maximum chlorophyll to carbon ratio – generic diazotrophs[l] |
| $\alpha_{G_F}$ | - | 0.01 | - | - | molC gChl$^{-1}$ (W m$^{-2}$)$^{-1}$ d$^{-1}$ | Generic diazotrophs, initial slope of photosynthetic response[a, d, g, k] |
| $m_{G_F}$ | - | 0.18 | - | - | d$^{-1}$ | Generic diazotrophs mortality rate[d] |
| $l_{G_F}$ | - | 0.05 | - | - | d$^{-1}$ | Generic diazotrophs respiration rate[d] |

| Model version: | H0 | H1 | H2 | H3 | Units | Description |
|---|---|---|---|---|---|---|
| $\mu_{U_F}^0$ | - | - | 0.25 | 0.25 | d$^{-1}$ | Reference unicellular cyanobacteria maximum growth rate at T = 0ºC |
| $k_{U_F}^{DIP}$ | - | - | 0.004 | 0.004 | mmol m$^{-3}$ | Unicellular cyanobacteria DIP uptake half-saturation |
| $\theta_{U_F}^{max}$ | - | - | 0.053 | 0.053 | mg Chl (mg C)$^{-1}$ | Maximum chlorophyll to carbon ratio – unicellular cyanobacteria |
| $\alpha_{U_F}$ | - | - | 0.05 | 0.05 | molC gChl$^{-1}$ (W m$^{-2}$)$^{-1}$ d$^{-1}$ | Unicellular cyanobacteria, initial slope of photosynthetic response |
| $m_{U_F}$ | - | - | 0.20 | 0.2 | d$^{-1}$ | Unicellular cyanobacteria mortality rate |
| $l_{U_F}$ | - | - | 0.05 | 0.05 | d$^{-1}$ | Unicellular cyanobacteria respiration rate |
| $g_{U_F}^{max}$ | - | - | 0.2 | 0.2 | d$^{-1}$ | Zooplankton maximum grazing rate on unicellular cyanobacteria |
| $k_{Zoo}^{U_F}$ | - | - | 0.001 | 0.001 | (mmol m$^{-3}$)$^2$ | Square zooplankton grazing half-saturation on unicellular cyanobacteria |
| $\mu_{C_F}^0$ | - | - | 0.25 | 0.25 | d$^{-1}$ | Reference colonial cyanobacteria maximum growth rate at T = 0ºC |
| $k_{C_F}^{DIP}$ | - | - | 0.004 | 0.004 | mmol m$^{-3}$ | Colonial cyanobacteria DIP uptake half-saturation |
| $\theta_{C_F}^{max}$ | - | - | 0.053 | 0.053 | mg Chl (mg C)$^{-1}$ | Maximum chlorophyll to carbon ratio – colonial cyanobacteria |
| $\alpha_{C_F}$ | - | - | 0.05 | 0.05 | molC gChl$^{-1}$ (W m$^{-2}$)$^{-1}$ d$^{-1}$ | Colonial cyanobacteria, initial slope of photosynthetic response |
| $m_{C_F}$ | - | - | 0.18 | 0.05 | d$^{-1}$ | Colonial cyanobacteria mortality rate |
| $l_{C_F}$ | - | - | 0.18 | 0.05 | d$^{-1}$ | Colonial cyanobacteria respiration rate |

| Model version: | H0 | H1 | H2 | H3 | Units | Description |
|---|---|---|---|---|---|---|
| $\mu_{H_F}^0$ | - | - | - | 0.2 | d$^{-1}$ | Reference heterotrophs maximum growth rate at T = 0ºC[m] |
| $k_{H_F}^{DIP}$ | - | - | - | 0.001 | mmol m$^{-3}$ | Heterotrophs DIP uptake half-saturation |
| $k_{H_F}^{DS}$ | - | - | - | 0.001 | mmol m$^{-3}$ | Heterotrophs organic phosphorus uptake half-saturation |
| $m_{H_F}$ | - | - | - | 0.2 | d$^{-1}$ | Heterotrophs mortality rate |
| $l_{H_F}$ | - | - | - | 0.05 | d$^{-1}$ | Heterotrophs respiration rate |

a. Moore, et al., (2004); b. Ward et al., (2013); c. Hood et al., (2001); d. Fennel et al., (2002); e. Capone et al., (1997); f. Berman-Frank et al., (2001); g. Hutchins et al., (2007); h. Kranz et al., (2010); i. Hong et al., (2017); k. Geider et al., (1997); l. Fennel et al., (2006); m. Pomeroy and Wiebe, (2001)

5  **Table 4 Root-mean-square-errors between observations and corresponding simulated variables. Observations between 2005 and 2010 were used during model calibrations (i.e., assimilated). Observations between 2011 and 2014 are used for independent model validations (non-assimilated).** Control versions H0' and H3' are not shown.

**0-100 m**

| | 2005 – 2010 (assimilated) | | | | 2011 – 2014 (non-assimilated) | | | |
|---|---|---|---|---|---|---|---|---|
| | NO$_3$ | PO$_4$ | CHL | O$_2$ | NO$_3$ | PO$_4$ | CHL | O$_2$ |
| H0 | 0.71 | 0.04 | 0.15 | 7.57 | 0.60 | 0.04 | 0.16 | 6.39 |
| H1 | 0.77 | 0.04 | 0.14 | 6.99 | 0.66 | 0.04 | 0.15 | 7.08 |
| H2 | 0.78 | 0.04 | 0.14 | 6.96 | 0.75 | 0.04 | 0.14 | 6.66 |
| H3 | 1.04 | 0.05 | 0.14 | 7.35 | 1.50 | 0.05 | 0.13 | 6.22 |
| H3a | 1.04 | 0.06 | 0.12 | 7.10 | 1.41 | 0.09 | 0.14 | 6.47 |
| H3b | 1.91 | 0.05 | 0.14 | 7.94 | 2.15 | 0.05 | 0.16 | 8.13 |
| H3c | 1.01 | 0.06 | 0.12 | 7.05 | 1.06 | 0.08 | 0.14 | 6.55 |
| H3d | 1.60 | 0.05 | 0.19 | 7.91 | 1.78 | 0.05 | 0.19 | 8.21 |

**100 – 600 m**

| | NO$_3$ | PO$_4$ | CHL | O$_2$ | NO$_3$ | PO$_4$ | CHL | O$_2$ |
|---|---|---|---|---|---|---|---|---|
| H0 | 1.53 | 0.05 | 0.08 | 15.54 | 2.26 | 0.06 | 0.06 | 17.34 |
| H1 | 1.43 | 0.05 | 0.07 | 15.09 | 2.02 | 0.05 | 0.06 | 21.70 |
| H2 | 1.29 | 0.05 | 0.07 | 14.17 | 1.56 | 0.04 | 0.06 | 17.57 |
| H3 | 1.05 | 0.05 | 0.07 | 13.28 | 0.89 | 0.05 | 0.06 | 10.03 |
| H3a | 1.12 | 0.05 | 0.07 | 13.42 | 0.93 | 0.07 | 0.06 | 11.98 |
| H3b | 1.26 | 0.10 | 0.07 | 18.29 | 1.51 | 0.14 | 0.06 | 29.99 |
| H3c | 1.14 | 0.05 | 0.07 | 14.37 | 1.18 | 0.05 | 0.06 | 16.16 |
| H3d | 1.41 | 0.11 | 0.14 | 18.39 | 1.85 | 0.14 | 0.13 | 30.38 |