# Peer review of "Modelling the biogeochemical effects of heterotrophic and autotrophic N2 fixation in the Gulf of Aqaba (Israel), Red Sea"

_Biogeosciences, 2017_

## Referee Comment (RC1) · FMM Monteiro (Referee) · 4 Apr 2018

The study by Kuhn et al. investigates the role of heterotrophic nitrogen fixers in the Gulf of Aqaba. To do so, they develop the first ocean model of heterotrophic nitrogen fixers in a 1D setting which also includes other main types of nitrogen fixers. They optimise their model parameters with time series observations of this region and then validate their model's results looking at different model's version (with/without heterotrophic N2 fixers, with/without explicit N2 fixation . . .) to look at the sensitivity of the different types of nitrogen fixers on the ocean biogeochemistry. They find that heterotrophic N2 fixers are key in representing observed concentration of nitrate and oxygen in the deep ocean.

[Figure]

This is an important study as the first model study to include heterotrophic nitrogen fixers, a model which has been carefully optimised and validated with an extensive time series. The paper is very well-written, clear, concise, and presents well-designed modelling experiments. I thus strongly encourage its publication. My only concern is on the conclusion in relation to N* as the model doesn't capture very well the observations (see main comments below).

Main comments

N* model reproduction While the model (H3 in particular) does a very good job at representing NO3 and O2 observed concentrations, the model seems to be quite off from the N* data. This is not well enough highlighted in the paper which currently presents heterotrophic nitrogen fixers being key on reproducing the N* values. Because the model (even H3) is not able to capture most of the observed variability in N*, I don't think the model results support well enough this conclusion. I wonder what could be missing in the model and if you have thought about the role of preferential P remineralisation on N*. I wrote a paper in 2012 investigating the role of preferential remineralisation of P on the distribution of N* in the North Atlantic. In that region, this mechanism is necessary to reproduce the observed sub-surface maximum of N*. Here your model seems to have too small values in PO4 and NO3 at depth. Would it possible to test in your model the effect of preferential remineralisation of P to see if that helps reproducing the observed N* variability? If not, at least mention it. In our 2012 model, preferential remineralisation of P helped to get higher concentration of P, enhanced nitrogen fixation and then resulted in higher N* value at depth (as N* increases after the remineralisation of diazotrophic matter).

Monteiro, F. M., & Follows, M. J. (2012). On nitrogen fixation and preferential remineralization of phosphorus. Geophysical Research Letters, 39(6).

Minor comments

P2, Line 1: Need to mention about atmospheric N sources

P3, Line 5: Need to justify in the introduction why the Gulf of Aqaba is an interesting region to study nitrogen fixation.

P3, Lines 21-23: Sentence not clear. Can you amend?

P5, Line 16: The model description is confusing on if/how H0 and H0' represent N2 fixation. Here you say "without explicit N2 fixation . . . and follows the model equations described in Fennel et al. (2006, 2013)", which would mean that there is a non explicit representation of N2 fixation in the model. If so, replace with "with non-explicit N2 fixation representation" and then describe briefly how it is done in Fennel's papers. But later on there are many references to "with no N2 fixation" (P5/Line 18, Figure 4, P10 Line 5, P14 Line 20, . . .). Can you amend accordingly?

P6, Line 5: Isn't there any evidence of DDA in this region or atmospheric source of N? Please add comments as they can be potential important sources of N.

P7, Line 2: I could not find how denitrification is represented in the model. Can you make sure it is described?

P9, Line 10: Need to comment on the model performance for PO4. Especially for 100-600 m where H3 matches well observations for NO3, still cannot get PO4 right.

P9, Line 11-12: Can you refer to Figure 3 here?

P9, Lines 20-24: This paragraph does not add much so I would be inclined to remove it.

P10, Line 9: It would be good to add a statement on why heterotrophic N2 fixers improve the representation of deep oxygen as a key result of your paper.

P10, Line 18-20: Need to comment on potential reasons why H3 accumulates NO3 over time.

P11, Line 3-8: Need to add comments on why H3 and H2 have much higher N2 fixation rate than observations between 0-DCM in the Summer 2010 .

P11, Line 17-18: I would be more subtle about the models' abilities to replicate N* as it is still quite far from the observations (see my main comments above).

P12, Lines 17-32: While this is an interesting section about the contribution of N2 fixation on PP, why include results from H3a which is not as realistic as H3?

P13, Section 5.3: One of the main points of the paper is to highlight the important role of heterotrophic nitrogen fixers. I feel this section could then be a lot stronger highlighting all the effect of heterotrophic N2 fixers on the ocean biogeochemistry of this region. Here for instance I would mention that heterotrophic N2 fixers improve the NO3 and O2 concentrations at depth, as well as the contribution of heterotrophic bacteria to total N2 fixation. Also, would it be possible to plot the model difference between H2 and H3 to show the impact of heterotrophic N2 fixation?

Figure 10 is not mentioned in the main text. It looks interesting so probably worse describing it at some point.

---

## Referee Comment (RC2) · Anonymous Referee #2 · 25 Apr 2018

GENERIC COMMENT:

This paper presents a model for heterotrophic nitrogen (N) fixation, implements it in a 1D context in the Arabian Sea and uses the model to test hypotheses on the relative contributions to N fixation by the different organisms. The subject fits perfectly in the Journal remits, and the work is highly relevant and it will be an important contribution to the topic of N fixation. I particularly liked the use of genetic algorithm for calibration and use of the model to test hypotheses. Authors set up a generally good framework to perform those test, unfortunately I believe that some further tests are needed in order to properly attribute the changes in the model outputs to the N fixation trait (see main comments).

MAIN COMMENTS: First of all, I would strongly encourage authors to be more comprehensive in the model description the supplement, because some key details are not clear, particularly on how the equations change in the different model set-up. In particular, in model H2 how the equation for zooplankton growth changes? Does Zooplankton see a single pool of phytoplankton formed by non-fixers and unicellular fixers, or does it graze separately on both? Given the non-linearity of the limitation function, the two options are very different. I would suggest writing explicitly all equations of H1, H2 and H3 that differ from H0 instead of summarising with sentences like "All other state variable equations are modified accordingly" The main concern is that authors directly compare models with very different structure and then attribute all changes observed in the results to the process without separating the impact of the biogeochemical process from the impact of the different model structure. For instance, in H3 authors added the heterotrophic nitrogen fixers, by adding to the implicit first-order mineralisation scheme of the small detritus, a more dynamic one that includes an explicit heterotrophic group (Hf). Such a big change in the model structure is bound to profoundly impact the model results, regardless of the N fixation ability of the heterotrophic group, because the whole dynamic of mineralisation is changed. I would recommend the authors to implement a H3' model where a non-fixer group of heterotrophic organisms uses organic and inorganic for of both N and P is used as Hf. The comparison between H0 and H3' would enable to understand how much of the mismatch between simulated and observed bottom waters N and O2 is due to an underestimation of mineralisation, while comparing H3' with H3 will allow to assess how the N fixation trait influence those dynamics. The comparison of H0 and H3' is much more important because the mineralisation rates have not been calibrated, and therefore could be affected by an initial bias. Similarly, when comparing model with 1 phytoplankton group (H0) with models with multiple PFT, all trophic dynamics can change, due to non-linearity in the grazing. Since Zooplankton dynamics are not shown, nor detailed equation for grazing and zooplankton growth in the different models, it is impossible to me to assess if the implementation of H1' and H2' similar to H3' are to be recommended or not.

Another main comment is related to the Redfieldian assumption. While I fully acknowledge the long tradition of Redfield ratio based models and data analyses, their power and their advantage, I'm always a bit concerned when these are used to draw conclusions on nutrient ratio dynamics, particularly in the short temporal and spatial scales. Phytoplankton internal nutrient ration and nutrient uptake are far from being constant and fixed to the Redfield ratio and they also varies a lot from species to species (e.g. Geider and La Roche, European Journal of Phycology, 2011, http://dx.doi.org/10.1017/S0967026201003456). I appreciate that this complexity is impossible to fully reproduce in biogeochemical model and therefore the Redfield assumption can still be used as first order approximation in simple biogeochemical model, however I would not use those model to analyse the instantaneous dynamic of nutrient ratios because this will be strongly affected by the huge assumption of fixed stoichiometry. Figure 6 itself shows how the model is not able to capture the wide variability of DIN:DIP ratio. For this reason, I would suggest to cut the part related to N*, or alternatively, repeat the analysis using annual means of DIN and DIP and include a discussion on the importance of non-Redfieldian dynamics.

SPECIFIC COMMENTS: Page 5, lines 19-20: in H0' N fixation and denitrification are balanced: where did denitrification occur? In the benthos? I recommend adding some detail to better interpret the vertical dynamics simulated by this model implementation Section 4.2.2.: Top left panel of figure 5 shows that model H2 and H3 are significantly overestimating surface nitrogen in the last 4 years of calibration, with the exception of the deep mixing events in winter 2007/2008. H3 largely overestimates surface nitrogen also in the validation period (figure 8). This important dynamic is not discussed in the paper. Section 4.2.4.: while I agree that H3 better compares with observed deep values, in the last couple of years a significant trend in deep nitrogen appears in the simulation and it's not in the data. I suggest authors to comment on that.

TECHNICAL COMMENT: Page 12, line 18: in 10b, the dot corresponding to Capone and Carpenter 1982 shows a N fixation equal or close to 0, that is quite different from the values simulated by the different flavour of H3 Figures 2,4,7: I

recommend the authors to redraw the picture using a perceptually uniform and colour-blind friendly colourmap like viridis, inferno, magma or plasma in Python or Parula in Matlab. More details on the importance of this in the following video https://www.youtube.com/watch?v=xAoljeRJ3lU

―――――――――――――――――――――

---

## Referee Comment (RC3) · Anonymous Referee #3 · 26 Apr 2018

The authors present a study which compares a series of 1-D biogeochemical models of increasing complexity with respect to the representation of different diazotrophic organisms against an in situ time series data set from a location in the Gulf of Aqaba in the Red Sea. Comparison of the output of these models with the in situ data and specifically the nutrient concentrations and ratios/differences between N and P concentrations (as quantified by the derived N* variable) is subsequently used to argue for a substantive contribution by heterotrophic diazotrophs to N2 fixation within the region. The manuscript is well written and in general the study rationale and model experiments appeared well designed (although see specific comments below). The subsequent results and potential implications of the study were certainly interesting and overall I felt there was much of value within the study. However, as outlined below,

[Figure]

I have a few concerns I would like to see the reviewers address.

Major comment:

The authors appear to undertake a thorough job of optimising many of the parameters related to their model (see e.g. Page 6 and Table 2). However, given the key question(s) being addressed within the study I was somewhat surprised that potential variability in what I would consider to be the key parameters in dictating how the diazotrophs interact with the N and P cycles were fixed, with no exploration of potential variability in these parameters. Specifically, the values of the N:P ratio within both the non-diazotrophs and the diazotrophs were fixed at 16:1 (Page 1 of Supplm.) and 45:1 (Page 5 of Supplm.) respectively. In contrast it is now fairly well recognised not only that N:P ratios within organic material can vary (see e.g. Martiny et al. 2013 Nature Geo. 6 279-283) but also (and crucially within the current context), that inferences of N2 fixation rates and interactions between diazotrophy and the cycling of N and P are highly dependent on both assumed values of these ratios and any variability within these (Mills and Arrigo 2010 Nature Geo. 4 412-416; Weber and Deutsch 2012 Nature 489 419-422). Consequently, I would suggest the authors should at least consider the implications of their assumed fixed N:P ratios for their interpretation and conclusions and perhaps also consider performing some sensitivity analysis around these currently fixed assumptions.

Specific comments:

Page 3, Line 2: It would be worthwhile directly stating the laboratory based studies considered here were specific to Trichodesmium. As far as I am aware we have little information on how other groups might be expected to respond to the drivers mentioned.

Page 5, Line 21: '. . . a generic autotrophic diazotroph. . .'

Page 6, Lines 10-25: I was unclear whether this parameter optimisation method was

performed for each of the models (H0 – H3 etc) independently or a single parameter set was used? Additionally, see major comment above, did the authors consider using the parameter optimization method for the non-diazoptroph and diazotroph N:P ratios? See also Page 14, Lines 11-16, I was unsure why this choice was made, it appears to be a big assumption within the current context.

Page 7, Line 13: maximum reported growth rates for Trichodesmium are actually >0.5 d-1, see Hong et al. (2017) Science 356 527-531

Page 8, Line 9: see major comment above. Either this assumption should be justified, or, preferably I would suggest, some effort could be made to perform a sensitivity analysis of how the assumption influences the results/conclusions.

Page 8, Line 14: again related to comments above, some speculation on how this happens would seem appropriate in the context of this study. As a suggestion, uptake at high N:P ratios by non-diazotrophs might be one potential mechanism for shifting from inputs of nutrients with an apparent 'excess' N (i.e. positive N*) to an apparent deficit (negative N*), see e.g. Mills and Arrigo (2010).

Page 9, Line 19 (also Page 12, Lines 7-8): it is notable that even the most complex model struggles to reproduce the observed range in N* and I wondered whether the restriction placed on the models through the assumption of the fixed N:P ratios may be responsible for this?

A final general point which is also related to many of those above, within the context of this study I felt that some of the important details relating to the model which were presented within the supplement would be more appropriately outlined within the main body of the text as they are likely fundamental to interpretation.

---

## Referee Comment (RC4) · Anonymous Referee #4 · 3 May 2018

General evaluation

This ms reports a 1D biogeochemical model analysis of time-series data from the Gulf od Aqaba from 2006–2014. The authors compare the behaviour of models with different diazotroph community structures representing various combinations of autotrophic and heterotrophic diazotrophs. While all model versions perform similarly with respect to surface chlorophyll, only models with diazotrophy can reproduce observed nutrient (N:P) ratios and heterotrophic diazotrophy is required to explain the vertical structure of nutrient and O2 concentrations.

In general, I find this study somewhat unconvincing. The model is overly simplistic in its mechanistic foundation and ignores processes I consider essential for this kind of analysis. While I do not dispute the potential importance of heterotrophic diazotrophy

for marine biogeochemistry, the conclusions and particularly the title appear overly optimistic and not well justified. The ms also appears to have been prepared rather sloppily and not thought through. The main problem is that all diazotroph parameters are unconstrained by the data, which, as outlined below, may be a consequence of the overly simplistic nature of the model or of an inappropriate cost function. Thus, in order to turn this ms into a useful contribution, the model or the cost function (or both) must be redesigned so as to achieve sensitivity to the diazotroph-related parameters.

Specific points

1. Starting with the title, I find the wording inappropriate. While it might be possible to obtain biogeochemical evidence from a model analysis, this is certainly not the case here. I would suggest something like "Modelling heterotrophic N2 fixation ..."

2. Model structure. Although the authors stress that they intended to analyse mechanistic assumptions (l. 15, p. 14), I find that the model is mechanistically rather weakly founded. While simplicity is of course an important goal in model development, one must take care not to over-simplify and neglect essential processes. I think this should be at least discussed thoroughly to put the results into the right perspective. The two assumptions I find most troubling are those of (1) constant (Redfield) stoichiometry of the autotrophs and (2) obligate diazotrophy, both of which are mechanistically wrong. Fernandez-Castro et al., J. Plank. Res. 38:946 (2016), FC in the following, applied a model with variable stoichiometry and facultative diazotrophy in the subtropical North Atlantic, where the vertical distribution of N, P, and N* poses similar difficulties as in the present ms. The model of FC is otherwise very similar in structure to the present one (phytoplankton, diazotrophs, zooplankton, detritus, nutrients, DOM), so I think the differences should be discussed, particularly with respect to the relations among stoichiometry, export and remineralisation.

Comparing the parameter settings between FC and the present model, I notice a very strong discrepancy (more than a factor of 10) in the initial-slope parameter (alpha) for

photosynthesis in diazotrophs, although the units are the same in both models. It is not clear from the ms how or why the very low alpha was chosen (no reference given and not optimised). But it appears to be an important parameter given that the analysis is about the vertical structure and alpha basically defines how deep in the water column autotrophic N2 fixation can occur.

Another parameter that appears rather low is the maximum growth rate of the autotrophic diazotrophs. For example, Holl & Montoya, J. Phycol. 44:929 (2008) reported growth rates greater than 0.6/d for Trichodesmium grown in a chemostat, so a maximum (actually potential) rate parameter of 0.25/d appears unrealistically low. My impression is that these low settings reduce diazotrophy too much, maybe just compensating for the assumption of obligate diazotrophy but maybe also being responsible for the requirement of aphotic N2 fixation in the present model.

Further, the authors say that the diazotroph parameters were unconstrained by the data and that the parameter setting were taken from the literature, but do not provide references in Table 3 or elsewhere. The ms also does not say how it was determined that the parameters were unconstrained by the data. This seems inappropriate to me, since this is specifically a model study about diazotrophy, so I expect that great care is taken to select appropriate parameter settings. The fact that the diazotroph parameters are unconstrained by the data makes the choice of data appear questionable to me. In my view, the data should be able to constrain the most important aspects of a model's performance, and if this is not the case, one should try to either find better data or develop a better cost function (see below). The problem is that the inability to constrain the model parameters with the data implies that the associated processes are actually irrelevant. The simple fact that the authors observe better model performance when including diazotrophs implies that the associated parameters must have an effect, so I expect that a better cost function can in fact be designed which is capable of constraining those paramenters.

3. Model evaluation. The authors report that they performed sensitivity analyses to
obtain information of sensitive model parameters but they do not say how the sensitivity was quantified nor present any results from the sensitivity analyses. This could well be done in the supplement, but it es important for those who want to work with the model later.

The authors mention that they considered the first year of the model simulations as spinup but do not say how the model was initialised (from observations? what about the non-observed variables?). From my own experience with 1D modelling, one year is a rather short period for a spinup. Did the authors try longer spinups in order to find out whether the model is sufficiently close to a quasi-steady-state after one year? This should be discussed as well. It is this kind of omission, together with missing entries in the list of references (e.g., Fernandez 2011 and Smith 1936), that leaves an impression of sloppiness.

4. Parameter estimation. The authors apply RMSEs of absolute concentrations to obtain a measure of model-data misfit. This cost function will not be sensitive to large relative deviations if the absolute concentrations are low. Thus, it is only logical that the inability of the model to reproduce the negative N* in the surface waters "is not a source of large data-model discrepancies" (l. 8, p. 12). Introducing relative-error information or local scaling into the cost function could help here. The most important shortcoming of the authors' cost function, however, is that it neglects error correlations, see, e.g., Schartau et al., Biogeosci. 14:1647 (2017).

5. Figures. The use of log-scales in Fig. 9 makes it impossible to see the differences among models and between models and data. Please use a linear scale.

6. Conclusions. As it stands, the conclusions are not sufficiently supported be the model analysis described. In particular, the conclusions about aphotic N2 fixation are compromised by the choice of unrealistic parameter values constraining autotrophic diazotrophy to the very surface. If inferences about heterotrophic diazotrophy are to be drawn, at least the parameters determining the depth distribution of autotrophic

diazotrophy must be analysed with a detailed sensitivity analysis. The current analysis cannot say whether the deep N signal is really due to aphotic N2 fixation or exported material from the surface.

---

## Author Comment (AC1) · 26 May 2018

Reviewer 1 has strongly encouraged our manuscript for publication and described it as "clear, concise, and present(ing) well-designed modelling experiments". We are grateful for the positive assessment and appreciate the constructive comments, which we respond to in more detail below.

(Responses to each comment are in bold text)

Main comments:

N* model reproduction While the model (H3 in particular) does a very good job at representing NO3 and O2 observed concentrations, the model seems to be quite off from the N* data. This is not well enough highlighted in the paper which currently

presents heterotrophic nitrogen fixers being key on reproducing the N* values. Because the model (even H3) is not able to capture most of the observed variability in N*, I don't think the model results support well enough this conclusion. I wonder what could be missing in the model and if you have thought about the role of preferential P remineralisation on N*. I wrote a paper in 2012 investigating the role of preferential remineralisation of P on the distribution of N* in the North Atlantic. In that region, this mechanism is necessary to reproduce the observed sub-surface maximum of N*. Here your model seems to have too small values in PO4 and NO3 at depth. Would it possible to test in your model the effect of preferential remineralisation of P to see if that helps re- producing the observed N* variability? If not, at least mention it. In our 2012 model, preferential remineralisation of P helped to get higher concentration of P, enhanced nitrogen fixation and then resulted in higher N* value at depth (as N* increases after the remineralisation of diazotrophic matter). Monteiro, F. M., Follows, M. J. (2012). On nitrogen fixation and preferential reminer- alization of phosphorus. Geophysical Research Letters, 39(6).

**Thank you for suggesting this paper. We will include the citation and acknowledge preferential demineralization of P as a mechanism acting to generate the observed N\* values. Nevertheless, as suggested by other reviewers, we will move our focus away from N\* by removing figures 3 and 6 from the manuscript.**

Minor comments P2, Line 1: Need to mention about atmospheric N sources

**Response: Taken. The sentence will be changed to:** *"Locally the supply of new nitrogen can occur through several mechanisms, including microbially mediated N2 fixation, diapycnal mixing injecting deep nitrate (NO3) into the surface, lateral transport, atmospheric N sources and riverine input."*

P3, Line 5: Need to justify in the introduction why the Gulf of Aqaba is an interesting region to study nitrogen fixation.

**Response: Taken. The referred paragraph in the introduction will be changed**

**to: "In this study we explore the biogeochemical signatures that result from different assumptions about the ecological niches occupied by diazotrophs. We use the Gulf of Aqaba, a northern extension of the Red Sea. Aside from the reported presence of diverse diazotrophs types, the morphology of the Gulf of Aqaba limits horizontal transport of deep waters, thus allowing us to simplify the physical complexity of the model and focus of the biological component."**

P3, Lines 21-23: Sentence not clear. Can you amend?

**Response: Taken. We will remove the phrase** *"(. . .) the Gulf does not have permanent vertical stratification (. . .)"* **for clarity. The modified sentence will read:** *"Since inflow is restricted to warm surface waters, the Gulf's deep water masses (>300 m) are locally formed (Wolf-Vecht et al., 1992; Biton et al., 2008) and have negligible horizontal transport toward the exterior (Klinker et al., 1976; Manasrah et al., 2006)."*

P5, Line 16: The model description is confusing on if/how H0 and H0' represent N2 fixation. Here you say "without explicit N2 fixation . . . and follows the model equations described in Fennel et al. (2006, 2013)", which would mean that there is a non explicit representation of N2 fixation in the model. If so, replace with "with non-explicit N2 fixation representation" and then describe briefly how it is done in Fennel's papers. But later on there are many references to "with no N2 fixation" (P5/Line 18, Figure 4, P10 Line 5, P14 Line 20, . . .). Can you amend accordingly?

**Response: We intend to modify the description as follows:** *"H0 is the base model without diazotrophic plankton groups and follows the model equations described in Fennel et al., (2006, 2013). We test this model with and without a sediment denitrification flux, denoted as H0 and H0', respectively. In other words, H0 includes denitrification but no N2 fixation, thus neglecting the importance of N2 fixation. H0' does include neither denitrification, nor N2 fixation, and thus the underlying assumption of this model version is that inputs from N2*

*fixation and losses of fixed nitrogen due to denitrification are balanced."*

P6, Line 5: Isn't there any evidence of DDA in this region or atmospheric source of N? Please add comments as they can be potential important sources of N.

**Response: Taken. Discussion will be added on other potential sources of nitrogen. Below are some initial thoughts:**

**a) DDA DNA has been detected in the region, but not as abundant as unicellular, Trichodesmium and proteobacteria (Kimor et al. 1992, Foster et al., 2009). In general, due to the oligotrophic characteristics of the Gulf of Aqaba, small phytoplankton species (<8 micrometers) contribute more than 90% of the chlorophyll-a standing stock (Lindell and Post 1995). Dinoflagellates and diatoms together correspond to less than 5% of the phytoplankton biomass, except during ephemeral diatom blooms during spring when they can account for nearly 50% of the total biomass (Al-Najjar et al., 2007).**

**b) Recently, it has been shown that atmospheric dust input does not correlate with chlorophyll variability in the surface waters of the Gulf of Aqaba (Torfstein and Kienast, 2018). Nevertheless, based on measurements of local aerosols composition and a dust deposition model, it is estimated that atmospheric deposition nitrogen flux could support over 10% of surface primary production (Chen et al., 2007). This estimate has a relatively large uncertainty due to errors associated with the deposition flux calculation and the temporal variability in dust flux (Chen et al., 2007). Moreover, very low N concentrations and lower than Redfield N:P ratios from the surface down to 80 m were observed during the same time period of Chen et al. study (Foster et al., 2009).**

**References:**

**Kimor, B., Gordon, N., Neori, A. (1992). Symbiotic associations among the microplankton in oligotrophic marine environments, with special reference to the**

Gulf of Aqaba, Red Sea. Journal of Plankton research, 14(9), 1217-1231.

Foster, R. A., Paytan, A., Zehr, J. P. (2009). Seasonality of N2 fixation and nifH gene diversity in the Gulf of Aqaba (Red Sea). Limnology and Oceanography, 54(1), 219-233.

Lindell, D., Post, A. F. (1995). Ultraphytoplankton succession is triggered by deep winter mixing in the Gulf of Aqaba (Eilat), Red Sea. Limnology and Oceanography, 40(6), 1130-1141.

Al-Najjar, T., Badran, M. I., Richter, C., Meyerhoefer, M., Sommer, U. (2007). Seasonal dynamics of phytoplankton in the Gulf of Aqaba, Red Sea. Hydrobiologia, 579(1), 69-83.

Torfstein, A., Kienast, S. S. (2018). No Correlation Between Atmospheric Dust and Surface Ocean Chlorophyll‐a in the Oligotrophic Gulf of Aqaba, Northern Red Sea. Journal of Geophysical Research: Biogeosciences, 123(2), 391-405.

Chen, Y., Mills, S., Street, J., Golan, D., Post, A., Jacobson, M., Paytan, A. (2007). Estimates of atmospheric dry deposition and associated input of nutrients to Gulf of Aqaba seawater. Journal of Geophysical Research: Atmospheres, 112(D4).

P7, Line 2: I could not find how denitrification is represented in the model. Can you make sure it is described?

**Response: The denitrification representation was included in the model description in the supplement:** *"When present, the denitrification flux follows Fennel et al. (2013) with a loss fraction 6 mol N2 per mol of organic matter remineralized".* **We will include this descriptive sentence in the main text for clarity.**

P9, Line 10: Need to comment on the model performance for PO4. Especially for 100-600 m where H3 matches well observations for NO3, still cannot get PO4 right.

**Response: We will expand the text to remark that models have a similar performance with respect to PO4, all underestimating total deep PO4 by the end of the series. This suggests that all models lack a process affecting PO4. As mentioned in Monteiro et al., (2012), processes affecting PO4, and not considered in this simplified model, may include horizontal physical transport in and out of the domain, phytoplankton stoichiometry, atmospheric deposition, and preferential PO4 remineralization. The latest may particularly affect PO4 at depth.**

P9, Line 11-12: Can you refer to Figure 3 here?

**Response: This is likely to change, as Figure 3 will be removed following the comments of other reviewers.**

P9, Lines 20-24: This paragraph does not add much so I would be inclined to remove it.

**Response: Taken. Note that Figure 6 (N\*) will also be removed.**

P10, Line 9: It would be good to add a statement on why heterotrophic N2 fixers improve the representation of deep oxygen as a key result of your paper.

**Response: Taken. We will explore deeper why this occurs and include it.**

P10, Line 18-20: Need to comment on potential reasons why H3 accumulates NO3 over time.

**Response: Taken. The revised version will include discussion on this matter. Our initial thought is that all models that consider nitrogen fixation accumulate nitrogen at different rates, as they enrich the nitrogen content of detritus, which is then remineralized at depth over time. The accumulation rate is also affected by the frequency of inter-annual deep-mixing events. An appropriate amount of N2 fixation is achieved with H3 with heterotrophic fixers. While similar amounts of fixed N could also be achieved by the autotrophic groups alone if their parameters were changed, this would require increased N2 fixation by autotrophic**

organisms and likely lead to unrealistic rates of N2 fixation in the surface. A full
validation and calibration of the simulated balance in contributions from each
diazotrophic group can only be achieved when more observational information
becomes available.

P11, Line 3-8: Need to add comments on why H3 and H2 have much higher N2 fixation
rate than observations between 0-DCM in the Summer 2010.

**Response: Taken. In H3 and H2 the large N2 fixation rates during summer are
due to the contribution of blooming in the simulated behavior of Trichodesmium.
This is likely to be overestimated under the current model configuration, as anec-
dotal evidence suggests that Trichodesmium blooms are more ephemeral and
do not have a predictable seasonal cycle. Unfortunately, we do not count with
routine sampling to be able to affirm the above. This model will require further
calibration as more information becomes available to verify the seasonal cycle
of nitrogen fixation in the region.**

P11, Line 17-18: I would be more subtle about the models' abilities to replicate N* as it
is still quite far from the observations (see my main comments above).

**Response: Following this and other reviewer's comments we have decided to
remove the figures and discussion concerning N*.**

P12, Lines 17-32: While this is an interesting section about the contribution of N2
fixation on PP, why include results from H3a which is not as realistic as H3? **Response:
Taken, will be removed.**

P13, Section 5.3: One of the main points of the paper is to highlight the important
role of heterotrophic nitrogen fixers. I feel this section could then be a lot stronger
highlighting all the effect of heterotrophic N2 fixers on the ocean biogeochemistry of
this region. Here for instance I would mention that heterotrophic N2 fixers improve
the NO3 and O2 concentrations at depth, as well as the contribution of heterotrophic

bacteria to total N2 fixation. Also, would it be possible to plot the model difference between H2 and H3 to show the impact of heterotrophic N2 fixation?

**Response: Taken. We will add more emphasis on the effects on ocean biochemistry and add a figure of the difference in N2 fixation between H2 and H3. Already, this difference, in terms of total N2 fixation, can be seen in Figure 10b.**

Figure 10 is not mentioned in the main text. It looks interesting so probably worse describing it at some point.

**Response: Thank you for pointing this out. The results in this figure are briefly mentioned in section 5.2 (How does N2 fixation Contribute to Primary Production?), however we did not referred specifically to the figure by mistake.**

---

## Author Comment (AC2) · 26 May 2018

(Reviews are included in regular font; Responses are in bold font)

This paper presents a model for heterotrophic nitrogen (N) fixation, implements it in a 1D context in the Arabian Sea and uses the model to test hypotheses on the relative contributions to N fixation by the different organisms. The subject fits perfectly in the Journal remits, and the work is highly relevant and it will be an important contribution to the topic of N fixation. I particularly liked the use of genetic algorithm for calibration and use of the model to test hypotheses. Authors set up a generally good framework to perform those test, unfortunately I believe that some further tests are needed in order to properly attribute the changes in the model outputs to the N fixation trait (see main

comments).

**Response: We are grateful for the positive assessment and appreciate the constructive comments, which we respond to in more detail below.**

MAIN COMMENTS: First of all, I would strongly encourage authors to be more comprehensive in the model description the supplement, because some key details are not clear, particularly on how the equations change in the different model set-up. In particular, in model H2 how the equation for zooplankton growth changes? Does Zooplankton see a single pool of phytoplankton formed by non-fixers and unicellular fixers, or does it graze separately on both? Given the non-linearity of the limitation function, the two options are very different. I would suggest writing explicitly all equations of H1, H2 and H3 that differ from H0 instead of summarising with sentences like "All other state variable equations are modified accordingly"

**Response: Taken. The complete sets of equations will be included and described in more detail to clarify the model assumptions.**

The main concern is that authors directly compare models with very different structure and then attribute all changes observed in the results to the process without separating the impact of the biogeochemical process from the impact of the different model structure. For instance, in H3 authors added the heterotrophic nitrogen fixers, by adding to the implicit first-order mineralisation scheme of the small detritus, a more dynamic one that includes an explicit heterotrophic group (Hf). Such a big change in the model structure is bound to profoundly impact the model results, regardless of the N fixation ability of the heterotrophic group, because the whole dynamic of mineralisation is changed. I would recommend the authors to implement a H3' model where a non-fixer group of heterotrophic organisms uses organic and inorganic for of both N and P is used as Hf. The comparison between H0 and H3' would enable to understand how much of the mismatch between simulated and observed bottom waters N and O2 is due to an underestimation of mineralisation, while comparing H3' with H3 will allow to

assess how the N fixation trait influence those dynamics. The comparison of H0 and H3' is much more important because the mineralisation rates have not been calibrated, and therefore could be affected by an initial bias.

**Response: This is a valuable point. Additional simulations and analysis will be performed as suggested.**

Similarly, when comparing model with 1 phytoplankton group (H0) with models with multiple PFT, all trophic dynamics can change, due to non-linearity in the graz- ing. Since Zooplankton dynamics are not shown, nor detailed equation for grazing and zooplankton growth in the different models, it is impossible to me to assess if the im- plementation of H1' and H2' similar to H3' are to be recommended or not.

**Response: As mentioned before, the supplement will be expanded to include all diazotrophs equations. In general, we have tried to keep the model structures as consistent and comparable as possible; however, we acknowledge that compar- ing models with different complexities is challenging due to the various changes that may occur as consequence of changing trophic dynamics.**

Another main comment is related to the Redfieldian assumption. While I fully ac- knowledge the long tradition of Redfield ratio based models and data analyses, their power and their advantage, I'm always a bit concerned when these are used to draw conclusions on nutrient ratio dynamics, particularly in the short temporal and spatial scales. Phytoplankton internal nutrient ration and nutrient uptake are far from being constant and fixed to the Redfield ratio and they also varies a lot from species to species (e.g. Geider and La Roche, European Journal of Phycology, 2011, http://dx.doi.org/10.1017/S0967026201003456). I appreciate that this complexity is impossible to fully reproduce in biogeochemical model and therefore the Redfield as- sumption can still be used as first order approximation in simple biogeochemical model, however I would not use those model to analyse the instantaneous dynamic of nutrient ratios because this will be strongly affected by the huge assumption of fixed stoichiometry. Figure 6 itself shows how the model is not able to capture the wide variability of DIN:DIP ratio. For this reason, I would suggest to cut the part related to N*, or alternatively, repeat the analysis using annual means of DIN and DIP and include a discussion on the importance of non-Redfieldian dynamics.

**Response: Taken. Following this and other reviewers comments we have taken the decision to remove figures and discussion concerning N*. We also will add sensitivity experiments to illustrate the effect of changes in diazotroph stoichiometry.**

SPECIFIC COMMENTS: Page 5, lines 19-20: in H0' N fixation and denitrification are balanced: where did denitrification occur? In the benthos? I recommend adding some detail to better interpret the vertical dynamics simulated by this model implementation

**Response: The denitrification representation was included in the model description in the supplement: "When present, the denitrification flux follows Fennel et al. (2013) with a loss fraction 6 mol N2 per mol of organic matter remineralized". We will include this descriptive sentence in the main text for clarity.**

Section 4.2.2.: Top left panel of figure 5 shows that model H2 and H3 are significantly overestimating surface nitrogen in the last 4 years of calibration, with the exception of the deep mixing events in winter 2007/2008. H3 largely overestimates surface nitrogen also in the validation period (figure 8). This important dynamic is not discussed in the paper.

**Response: We will be happy to further discuss this point in our revised manuscript. In general, geomorphology and bathymetry limit water flux exchange between the Gulf of Aqaba and the Red Sea to the upper 300 m. It this unlikely that horizontal transport could explain the observed accumulation in deep NO3, but exchange of surface waters during summer months does occur. A possible explanation is that some of the DIP produced should be exported to the outside of the Gulf, which our 1D model does not account for. Alternatively,**

**the contribution to N2 fixation of each type of diazotroph requires further validation, and possible mechanisms for limiting N2 fixation by heterotrophs at the surface may be needed.**

Section 4.2.4.: while I agree that H3 better compares with observed deep values, in the last couple of years a significant trend in deep nitrogen appears in the simulation and it's not in the data. I suggest authors to comment on that.

**Response: Please see previous response concerning export of DIP to outside the Gulf.**

TECHNICAL COMMENT: Page 12, line 18: in 10b, the dot corresponding to Capone and Carpenter 1982 shows a N fixation equal or close to 0, that is quite different from the values simulated by the different flavour of H3 Figures 2,4,7: I recommend the authors to redraw the picture using a perceptually uniform and colour-blind friendly colourmap like viridis, inferno, magma or plasma in Python or Parula in Matlab. More details on the importance of this in the following video https://www.youtube.com/watch?v=xAoljeRJ3lU

**Response: Taken, we will happily redraw with a color-blind friendly colormap.**

---

## Author Comment (AC3) · 26 May 2018

(Reviews are included in regular font; Responses are in bold font)

The authors present a study which compares a series of 1-D biogeochemical models of increasing complexity with respect to the representation of different diazotrophic organisms against an in situ time series data set from a location in the Gulf of Aqaba in the Red Sea. Comparison of the output of these models with the in situ data and specifically the nutrient concentrations and ratios/differences between N and P concentrations (as quantified by the derived N* variable) is subsequently used to argue for a substantive contribution by heterotrophic diazotrophs to N2 fixation within the region. The manuscript is well written and in general the study rationale and model

experiments appeared well designed (although see specific comments below). The subsequent results and potential implications of the study were certainly interesting and overall I felt there was much of value within the study. However, as outlined below, I have a few concerns I would like to see the reviewers address.

**Response: We are grateful for the overall positive assessment and appreciate the constructive comments, which we respond to in more detail below.**

Major comment:

The authors appear to undertake a thorough job of optimising many of the parameters related to their model (see e.g. Page 6 and Table 2). However, given the key question(s) being addressed within the study I was somewhat surprised that potential variability in what I would consider to be the key parameters in dictating how the diazotrophs interact with the N and P cycles were fixed, with no exploration of potential variability in these parameters. Specifically, the values of the N:P ratio within both the non-diazotrophs and the diazotrophs were fixed at 16:1 (Page 1 of Supplm.) and 45:1 (Page 5 of Supplm.) respectively. In contrast it is now fairly well recognised not only that N:P ratios within organic material can vary (see e.g. Martiny et al. 2013 Nature Geo. 6 279-283) but also (and crucially within the current context), that inferences of N2 fixation rates and interactions between diazotrophy and the cycling of N and P are highly dependent on both assumed values of these ratios and any variability within these (Mills and Arrigo 2010 Nature Geo. 4 412-416; Weber and Deutsch 2012 Nature 489 419-422). Consequently, I would suggest the authors should at least consider the implications of their assumed fixed N:P ratios for their interpretation and conclusions and perhaps also consider performing some sensitivity analysis around these currently fixed assumptions.

**Response: This is a very valuable point. We will perform the sensitivity analysis suggested and include the results either on the main manuscript or as part of the supplement. Results of the sensitivity analysis will be addressed and discussed**

**within the manuscript.**

Specific comments:

Page 3, Line 2: It would be worthwhile directly stating the laboratory based studies considered here were specific to Trichodesmium. As far as I am aware we have little information on how other groups might be expected to respond to the drivers mentioned.

**Response: Agree, the laboratory experiments cited consider only Trichodesmium, while the model study of Dutkiewicz et al. 2015 considers a set of generic diazotrophic organisms with various sizes and growth rates. The following line will be added: "To our knowledge, these laboratory experiments have only explored the reaction of Trichodesmium and less information is available about the effects of climate trends on other diazotrophic organisms"**

Page 5, Line 21: '. . . a generic autotrophic diazotroph. . .'

**Response: Thank you for pointing out this typo.**

Page 6, Lines 10-25: I was unclear whether this parameter optimisation method was performed for each of the models (H0 – H3 etc) independently or a single parameter set was used? Additionally, see major comment above, did the authors consider using the parameter optimization method for the non-diazoptroph and diazotroph N:P ratios? See also Page 14, Lines 11-16, I was unsure why this choice was made, it appears to be a big assumption within the current context.

**Response: Thank you for pointing out this issue with the clarity of the description of our methods. This part of the methods is described in section 3.3.1 (Optimized Parameters), however it may be more useful to introduce it before describing the optimization method. We will happily address it in the revised version.**

Page 7, Line 13: maximum reported growth rates for Trichodesmium are actually >0.5 d-1, see Hong et al. (2017) Science 356 527-531

**Response: Thank you for pointing us to this updated reference. We will modify the text accordingly in the revised manuscript.**

Page 8, Line 9: see major comment above. Either this assumption should be justified, or, preferably I would suggest, some effort could be made to perform a sensitivity analysis of how the assumption influences the results/conclusions.

**Response: Agree, see response to major comment above.**

Page 8, Line 14: again related to comments above, some speculation on how this happens would seem appropriate in the context of this study. As a suggestion, uptake at high N:P ratios by non-diazotrophs might be one potential mechanism for shifting from inputs of nutrients with an apparent 'excess' N (i.e. positive N*) to an apparent deficit (negative N*), see e.g. Mills and Arrigo (2010).

**Response: We will happily include your suggestion as we also consider this to be a likely mechanism.**

Page 9, Line 19 (also Page 12, Lines 7-8): it is notable that even the most complex model struggles to reproduce the observed range in N* and I wondered whether the restriction placed on the models through the assumption of the fixed N:P ratios may be responsible for this?

**Response: That is indeed a possibility that we have not explored. As brought up by Reviewer 1, other alternatives include horizontal physical transport in and out of the domain, phytoplankton stoichiometry, atmospheric deposition, and preferential PO4 remineralization. The latest may particularly affect PO4 at depth, while transport may affect surface values during summer (when exchange of surface waters with exterior waters has been reported to occur). Nevertheless, given the reservations about our discussion of N* that were expressed by several of the Reviewers we have decided to remove this figure and other references to N* in the text.**

A final general point which is also related to many of those above, within the context of this study I felt that some of the important details relating to the model which were presented within the supplement would be more appropriately outlined within the main body of the text as they are likely fundamental to interpretation.

**Response: Agree, we will expand the model description on the main text to include details such as the denitrification process brought up by Reviewers 1 and 2. Also, as requested by Reviewer 2, we will further expand the supplement to include all model equations explicitly. However, we consider that including the complete model description in the main text may make it excessively long and diverge the attention from the results.**

---

## Author Comment (AC4) · 27 May 2018

Response to Comments by Reviewer 4 (Reviews are included in regular font; Responses are in bold font)

General evaluation This ms reports a 1D biogeochemical model analysis of time-series data from the Gulf od Aqaba from 2006–2014. The authors compare the behaviour of models with different diazotroph community structures representing various combinations of autotrophic and heterotrophic diazotrophs. While all model versions perform similarly with respect to surface chlorophyll, only models with diazotrophy can reproduce observed nutrient (N:P) ratios and heterotrophic diazotrophy is required to explain the vertical structure of nutrient and O2 concentrations.

[Figure]

In general, I find this study somewhat unconvincing. The model is overly simplistic in its mechanistic foundation and ignores processes I consider essential for this kind of analysis. While I do not dispute the potential importance of heterotrophic diazotrophy for marine biogeochemistry, the conclusions and particularly the title appear overly optimistic and not well justified. The ms also appears to have been prepared rather sloppily and not thought through. The main problem is that all diazotroph parameters are unconstrained by the data, which, as outlined below, may be a consequence of the overly simplistic nature of the model or of an inappropriate cost function. Thus, in order to turn this ms into a useful contribution, the model or the cost function (or both) must be redesigned so as to achieve sensitivity to the diazotroph-related parameters.

**Response: We respond one-by-one to each of the critical points that the Reviewer raises:**

**With regard to the comment that "the model is overly simplistic," we would like to state that our analysis is exploratory and focused on the influence of N2 fixation, hence complexity in other parts of the ecosystem, although necessary in any global application, is intentionally minimized here. The Gulf of Aqaba is an oligotrophic system and our relatively simple model structure is able to capture the observed variability. With regard to the title being "overly optimistic and not well justified," we are happy to change the title according to this Reviewer's suggestion (also see response to detailed comment below).**

**With regard to the manuscript being "prepared rather sloppily and not thought through," we would like to point out that some of the information the Reviewer is requesting (i.e. literature sources of parameter values) is actually provided in the manuscript (Table 3) and discussed in the text for diazotrophic organisms (section). In our revision we will incorporate the additional information that the Reviewer is requesting, i.e. more details on the sensitivity experiments, initial conditions, more details on the ranges and literature sources for diazotrophic organisms.**

With regard to "diazotroph parameters are unconstrained by the data," we would point out that this is not an issue with the model or the cost function. It is generally accepted among modellers that measured bulk properties like chlorophyll, nutrients and oxygen do not constrain most rates including rates of grazing, phytoplankton mortality and, in our case, N2 fixation (see, for example, discussion about this topic on Ward et al., 2010). No redesign of the cost function or the model will change the fact that the measured properties in the Gulf of Aqaba do not directly constrain N2 fixation rates, simply because that information is not contained in the observations. The observations capture the impact that N2 fixation only indirectly through its influence on deep-water nutrient ratios.

Ward, B.A., Friedrichs, M.A.M., Anderson, T.R., Oschlies, A., 2010. Parameter optimization techniques and the problem of underdetermination in marine biogeochemical models. J. Mar. Syst. 81, 34–43.

Specific points 1. Starting with the title, I find the wording inappropriate. While it might be possible to obtain biogeochemical evidence from a model analysis, this is certainly not the case here. I would suggest something like "Modelling heterotrophic N2 fixation ..."

**Response: We will be happy to change that title as the Reviewer suggests. A tentative alternative is: "Modeling the biogeochemical effects of heterotrophic and autotrophic N2 fixation in the Gulf of Aqaba, Red Sea"**

2. Model structure. Although the authors stress that they intended to analyse mechanistic assumptions (l. 15, p. 14), I find that the model is mechanistically rather weakly founded. While simplicity is of course an important goal in model development, one must take care not to over-simplify and neglect essential processes. I think this should be at least discussed thoroughly to put the results into the right perspective. The two assumptions I find most troubling are those of (1) constant (Redfield) stoichiometry of the autotrophs and (2) obligate diazotrophy, both of which are mechanistically wrong.

Fernandez-Castro et al., J. Plank. Res. 38:946 (2016), FC in the following, applied a model with variable stoichiometry and facultative diazotrophy in the subtropical North Atlantic, where the vertical distribution of N, P, and N* poses similar difficulties as in the present ms. The model of FC is otherwise very similar in structure to the present one (phytoplankton, diazotrophs, zooplankton, detritus, nutrients, DOM), so I think the differences should be discussed, particularly with respect to the relations among stoichiometry, export and remineralisation.

**Response: We will be happy to further discuss the implications of our model assumptions and compare our model and results to those of Fernandez-Castro et al. in our revised manuscript. With regard to the particular assumption mentioned by the Reviewer, namely constant (Redfield) stoichiometry of the autotrophs and obligate diazotrophy, we would like to comment that the overwhelming majority of models (including those used in IPCC projections) assume constant stoichiometry and, where diazotrophy is explicitly included, obligate diazotrophy. In our revision we will include additional sensitivity experiments to show the implications of variable stoichiometry of the diazotrophs. The question of obligate diazotrophy may be boil down to semantics because autotrophic diazotrophs that aren't fixing N2 would behave like a non-diazotrophic phytoplankton, which are included in our model. We would also like to comment that our focus is not on the physiology of diazotrophs, which is analyzed in more detail by Fernandez-Castro et al. and citations therein. We acknowledge that modeling diazotrophic cells N allocation mechanisms is important to understand how they are diazotrophs are able to fixate N2 under conditions that are traditionally thought to limit the process. We will happily bring attention to this previously missing piece of information in our introduction and discussion.**

Comparing the parameter settings between FC and the present model, I notice a very strong discrepancy (more than a factor of 10) in the initial-slope parameter (alpha) for photosynthesis in diazotrophs, although the units are the same in both models. It is not

clear from the ms how or why the very low alpha was chosen (no reference given and not optimised). But it appears to be an important parameter given that the analysis is about the vertical structure and alpha basically defines how deep in the water column autotrophic N2 fixation can occur.

**Response: Our values of alpha are within the range of values typically used in ecosystem models with similar formulations (see Doney et al. 1996; Fennel et al., 2001; Fennel et al., 2002; Schartau and Oschlies 2003; Fennel et al., 2006; Moore et al., 2004). Please also notice that while FC's our alpha parameters share the same notation, they may not refer to the same parameter because our light limitation formulations are not the same. It is not straightforward to compare these values directly to each other. We will be happy to further discuss these considerations in our revised manuscript. Nevertheless, we would also like to note that FC's alpha values were subjectively adjusted from the original model configuration by Pahlow et al., 2013. FC report these parameters were adjusted to reduce the depth of N2 fixation in their model results and even then, they obtain significant differences between the simulated and observed vertical structure of N2 fixation. To us, these differences in parameter values simply exemplify the fact that transferring parameters measured in laboratory cultures to mathematical models that represent the real ocean, and/or transporting parameters from one model to another is challenging and corresponds to a different discussion in itself.**

**Doney, S., D. M. Glover, R. G., Najjars, 1996. A new coupled, one-dimensional biological-physical model for the upper ocean: Applications to the JGOFS Bermuda Atlantic Time- series Study (BATS) site. Deep-Sea Research II, Vol 43, No. 2-3 pp. 591-624.**

**Fennel, K., M. Losch, J. Schröter and M. Wenzel, 2001. Testing a marine ecosystem model: Sensitivity analysis and parameter optimization. Journal of Marine Systems 28/1-2, p.45-63**

Fennel, K., Spitz, Y.H., Letelier, R., Abbott, M.R., 2002. A deterministic model for N2 fixation at stn. ALOHA in the subtropical North Pacific Ocean. Deep-Sea Res. II 49, 149–174.

Fennel, K., Wilkin, J., Levin, J., Moisan, J., O'Reilly, J.E., Haidvogel, D., 2006. Nitrogen cycling in the Middle Atlantic Bight: Results from a three-dimensional model and implications for the North Atlantic nitrogen budget. Glob. Biogeochem. Cycles 20, 14. doi:10.1029/2005GB00245.

Schartau, M., Oschlies, A., 2003. Simultaneous data-based optimization of a 1D-ecosystem model at three locations in the North Atlantic: Part I - Method and parameter estimates. J. Mar. Res. 61, 765–793.

Moore, J.K., Doney, S., Lindsay, K., 2004. Upper ocean ecosystem dynamics and iron cycling in a global three-dimensional model. Glob. Biogeochem. Cycles 18.

Another parameter that appears rather low is the maximum growth rate of the autotrophic diazotrophs. For example, Holl Montoya, J. Phycol. 44:929 (2008) re- ported growth rates greater than 0.6/d for Trichodesmium grown in a chemostat, so a maximum (actually potential) rate parameter of 0.25/d appears unrealistically low. My impression is that these low settings reduce diazotrophy too much, maybe just compensating for the assumption of obligate diazotrophy but maybe also being responsible for the requirement of aphotic N2 fixation in the present model.

**Response: Please note that the maximum growth rate in the model is temperature dependent and that 0.25/d is the reference value at 0 degrees C (as stated in Table 3). At typical water temperatures in the Gulf of Aqaba of 20 degrees C this results in an actual maximum growth rate of 0.97/d, close to the value mentioned by the Reviewer.**

Further, the authors say that the diazotroph parameters were unconstrained by the data and that the parameter setting were taken from the literature, but do not provide

references in Table 3 or elsewhere. The ms also does not say how it was determined that the parameters were unconstrained by the data. This seems inappropriate to me, since this is specifically a model study about diazotrophy, so I expect that great care is taken to select appropriate parameter settings. The fact that the diazotroph parameters are unconstrained by the data makes the choice of data appear questionable to me. In my view, the data should be able to constrain the most important aspects of a model's performance, and if this is not the case, one should try to either find better data or develop a better cost function (see below). The problem is that the inability to constrain the model parameters with the data implies that the associated processes are actually irrelevant. The simple fact that the authors observe better model performance when including diazotrophs implies that the associated parameters must have an effect, so I expect that a better cost function can in fact be designed which is capable of constraining those paramenters.

**Response: With regard to the Reviewer's assertion that "parameter setting were taken from the literature, but do not provide references in Table 3 or elsewhere," we would like to point to Table 2 which lists ranges for each parameter from the published literature with the corresponding references (see columns 3 and 6).**

**In response to the comment that we do "not say how it was determined that the parameters were unconstrained by the data": Showing that measurements of chlorophyll, nutrients and oxygen do not constrain N2 fixation rates is not within the intended scope of this paper. In order to clarify, what we here mean by "unconstrained" is that, for example, chlorophyll measurements alone cannot provide any information about how much chlorophyll is due to diazotrophs and how much is due to non-diazotrophs. Therefore, this variable alone is unable to help in the determination of diazotrophic parameters (i.e., the parameters are unconstrained). The systematic calibration method we use relies on using direct observational counterparts (i.e., from the same location at least) to compare to the model output. From previous knowledge and experience using optimization**

**methods, we understand that we would need comprehensive N2 fixation and/or size-structured diazotrophs biomass data in order to constrain all diazotrophs parameters.**

**With regard to the comment that "the choice of data appear questionable to" the Reviewer, we would like to respond that we used all the data that was available. The suite of available measurements is very typical of multi-year oceanographic time series (most aren't as comprehensive and well-funded as the HOT and BATS programs).**

**With regard to the cost function we refer to the last paragraph in our response to the first comment.**

3. Model evaluation. The authors report that they performed sensitivity analyses to obtain information of sensitive model parameters but they do not say how the sensitivity was quantified nor present any results from the sensitivity analyses. This could well be done in the supplement, but it es important for those who want to work with the model later.

**Response: We will be happy to include that information in the Supplement.**

The authors mention that they considered the first year of the model simulations as spinup but do not say how the model was initialised (from observations? what about the non-observed variables?). From my own experience with 1D modelling, one year is a rather short period for a spinup. Did the authors try longer spinups in order to find out whether the model is sufficiently close to a quasi-steady-state after one year? This should be discussed as well. It is this kind of omission, together with missing entries in the list of references (e.g., Fernandez 2011 and Smith 1936), that leaves an impression of sloppiness.

**Response: We will be happy to include this additional information in the revised manuscript.**

4. Parameter estimation. The authors apply RMSEs of absolute concentrations to obtain a measure of model-data misfit. This cost function will not be sensitive to large relative deviations if the absolute concentrations are low. Thus, it is only logical that the inability of the model to reproduce the negative N* in the surface waters "is not a source of large data-model discrepancies" (l. 8, p. 12). Introducing relative-error information or local scaling into the cost function could help here. The most important shortcoming of the authors' cost function, however, is that it neglects error correlations, see, e.g., Schartau et al., Biogeosci. 14:1647 (2017).

**Response: Our cost function does indeed scale depending on the data type by weighing the contributions from the different variables by their standard deviations.**

5. Figures. The use of log-scales in Fig. 9 makes it impossible to see the differences among models and between models and data. Please use a linear scale.

**Response: We will be happy to provide the plots on a linear scale to illustrate why we have chosen the log-scale.**

6. Conclusions. As it stands, the conclusions are not sufficiently supported be the model analysis described. In particular, the conclusions about aphotic N2 fixation are compromised by the choice of unrealistic parameter values constraining autotrophic diazotrophy to the very surface. If inferences about heterotrophic diazotrophy are to be drawn, at least the parameters determining the depth distribution of autotrophic diazotrophy must be analysed with a detailed sensitivity analysis. The current analysis cannot say whether the deep N signal is really due to aphotic N2 fixation or exported material from the surface.

**Response: As per our responses above, the parameters for autotrophic diazotrophs are fully consistent with the existing literature.**

---

## Author Response (AR1)

**1. DETAILED RESPONSES TO REVIEWERS**

**Response to Comments by Reviewer 1 (FMM Monteiro)**

(Reviews are included in black font; Responses are in blue font)

The study by Kuhn et al. investigates the role of heterotrophic nitrogen fixers in the Gulf of Aqaba. To do so, they develop the first ocean model of heterotrophic nitrogen fixers in a 1D setting which also includes other main types of nitrogen fixers. They optimise their model parameters with time series observations of this region and then validate their model's results looking at different model's version (with/without heterotrophic N2 fixers, with/without explicit N2 fixation . . .) to look at the sensitivity of the different types of nitrogen fixers on the ocean biogeochemistry. They find that heterotrophic N2 fixers are key in representing observed concentration of nitrate and oxygen in the deep ocean.

This is an important study as the first model study to include heterotrophic nitrogen fixers, a model which has been carefully optimised and validated with an extensive time series. The paper is very well-written, clear, concise, and presents well-designed modelling experiments. I thus strongly encourage its publication. My only concern is on the conclusion in relation to N\* as the model doesn't capture very well the observations (see main comments below).

**Response:** We are grateful for the positive assessment and appreciate the constructive comments. As described in our detailed responses below, we have modified the manuscript accordingly and believe it has improved significantly as a result.

**Main comments**

N\* model reproduction While the model (H3 in particular) does a very good job at representing NO3 and O2 observed concentrations, the model seems to be quite off from the N\* data. This is not well enough highlighted in the paper which currently presents heterotrophic nitrogen fixers being key on reproducing the N\* values. Because the model (even H3) is not able to capture most of the observed variability in N\*, I don't think the model results support well enough this conclusion. I wonder what could be missing in the model and if you have thought about the role of preferential P remineralisation on N\*. I wrote a paper in 2012 investigating the role of preferential remineralisation of P on the distribution of N\* in the North Atlantic. In that region, this mechanism

is necessary to reproduce the observed sub-surface maximum of N\*. Here your model seems to have too small values in PO4 and NO3 at depth. Would it possible to test in your model the effect of preferential remineralisation of P to see if that helps re- producing the observed N\* variability? If not, at least mention it. In our 2012 model, preferential remineralisation of P helped to get higher concentration of P, enhanced nitrogen fixation and then resulted in higher N\* value at depth (as N\* increases after the remineralisation of diazotrophic matter).

Monteiro, F. M., & Follows, M. J. (2012). On nitrogen fixation and preferential reminer- alization of phosphorus. Geophysical Research Letters, 39(6).

**Response:** Thank you for suggesting this paper. We included this citation in section 5.4, where we discuss uncertainties and limitations of our study:

"...preferential PO4 remineralization may more directly increase deep-PO4 concentrations as organic matter is decomposed in the bottom layers. The inclusion of this process in a model of the North Atlantic Ocean improved the representations of biogeochemical characteristics of the area and increased the  $N_s$  fixation rates obtained by the model (Monteiro and Follows, 2012)."

Nevertheless, as suggested by other reviewers, we moved our focus away from N\* by removing figures 3 and 6 from the manuscript.

Minor

comments

P2, Line 1: Need to mention about atmospheric N sources

Response: We agree. The sentence was changed to:

"Locally the supply of new nitrogen can occur through several mechanisms, including microbially mediated  $N_2$  fixation, diapycnal mixing injecting deep nitrate (NO3) into the surface, lateral transport, atmospheric N sources and riverine input."

P3, Line 5: Need to justify in the introduction why the Gulf of Aqaba is an interesting region to study nitrogen fixation.

Response: Agree. The referred paragraph in the introduction was changed to:

"In this study we explore the biogeochemical signatures that result from different assumptions about the ecological niches occupied by diazotrophs. Our study area is the Gulf of Aqaba, a northern extension of the Red Sea. Aside from the reported presence of diverse diazotrophs types.

the morphology of the Gulf of Aqaba limits horizontal transport of deep waters, thus allowing us to simplify the physical model's complexity of the model and focus on the biological component."

P3, Lines 21-23: Sentence not clear. Can you amend?

Response: We simplified this sentence for clarity". The modified sentence reads:

"Since inflow is restricted to warm surface waters, the Gulf's deep water masses (>300 m) are locally formed (Wolf-Vecht et al., 1992; Biton et al., 2008) and have negligible horizontal transport toward the exterior (Klinker et al., 1976; Manasrah et al., 2006)."

P5, Line 16: The model description is confusing on if/how H0 and H0' represent N2 fixation. Here you say "without explicit N2 fixation . . . and follows the model equations described in Fennel et al. (2006, 2013)", which would mean that there is a non explicit representation of N2 fixation in the model. If so, replace with "with non-explicit N2 fixation representation" and then describe briefly how it is done in Fennel's papers. But later on there are many references to "with no N2 fixation" (P5/Line 18, Figure 4, P10 Line 5, P14 Line 20, . . .). Can you amend accordingly?

**Response:** We modified the description as follows:**

"H0 is the base model without diazotrophic plankton groups and follows the model equations described in Fennel et al., (2006, 2013). In general, the model follows Monod kinetics using a fixed N:P ratio ( $R_{N:P}^{nf}$ = 16). Sensitivity to the constant N:P ratios is explored in section 4.2.3. We test the H0 model with and without a sediment denitrification flux (model versions H0 and H0', respectively). H0 includes denitrification but no N2 fixation, as no diazotrophs are considered. H0' does include neither denitrification, nor diazotrophic organisms, and thus the underlying assumption of this model version is that there is a balance between inputs from N2 fixation and losses of fixed nitrogen due to denitrification. When present, the denitrification flux follows Fennel et al. (2013) with a loss fraction 6 mol N2 per mol of organic matter remineralized at the sediment-water interface. This generates an average sediment denitrification flux of 0.25 ± 0.46 mmol N m2 d4, with a maximum value of 3.01 mmol N m2 d4."

P6, Line 5: Isn't there any evidence of DDA in this region or atmospheric source of N? Please add comments as they can be potential important sources of N.

**Response:** We extended the discussion about other potential sources of nitrogen (section 5.4):

"There are other sources of nitrogen that were not explored in the present study and we discuss

here briefly. For instance, we did not include contributions to N2 fixation by diatom-diazotroph associations, which are significant in other regions. While the genetic material of diatomdiazotroph associations has been detected in the Gulf of Aqaba, it is not as abundant as unicellular diazotrophs, Trichodesmium and proteobacteria (Kimor et al. 1992, Foster et al., 2009). In general, due to the oligotrophic characteristics of the region, small phytoplankton species (<8 µm) contribute more than 90% of the chlorophyll-a standing stock (Lindell & Post 1995, Yahel et al. 1998). Dinoflagellates and diatoms together correspond to less than 5% of the phytoplankton biomass, except during ephemeral diatom blooms during spring when they can account for nearly 50% of the total biomass (Al-Najjar et al., 2006).

Another source of nitrogen that has received interest in this region is atmospheric deposition, as the Gulf receives considerable dust input from the surrounding deserts. Recently, it has been shown that atmospheric dust input does not correlate with chlorophyll variability in surface waters of the Gulf of Aqaba (Torfstein and Kienast, 2018). A previous study suggested that atmospheric deposition of nitrogen could support over 10% of surface primary production in the region, based on measurements of local aerosol composition and a dust deposition model (Chen et al., 2007). However, this estimate had a relatively large uncertainty due to errors associated with the deposition flux calculation and the temporal variability in dust flux (Chen et al., 2007). Moreover, very low nitrogen concentrations and N:P ratios lower than Redfield from the surface down to 80 m were observed during the same time period (Foster et al., 2009). Therefore, the role of atmospheric nitrogen inputs remains uncertain."

P7, Line 2: I could not find how denitrification is represented in the model. Can you make sure it is described?

**Response:** The denitrification representation was described on the supplement. Now we have included it in the methods sections:

"When present, the denitrification flux follows Fennel et al. (2013) with a loss fraction of 6 mol  $N_2$ per mol of organic matter remineralized at the sediment-water inetrface. This generates an average sediment denitrification flux of 0.25 ± 0.46 mmol N m2d4, with a maximum value of 3.01 mmol N m  ${}^{2}d^{4}$ ."

P9, Line 10: Need to comment on the model performance for PO4. Especially for 100-600 m where H3 matches well observations for NO3, still cannot get PO4 right.

**Response:** We now state more clearly that: "All model versions represent similar vertical distributions of  $PO_4$  and underestimate its deep-water concentrations by the end of the series" and we discuss this in the uncertainties and limitation section.

P9, Line 11-12: Can you refer to Figure 3 here?

**Response:** The previous Figure 3 was removed. Figure 3 now refers to a different result.

P9, Lines 20-24: This paragraph does not add much so I would be inclined to remove it.

**Response:** This paragraph was deleted.

P10, Line 9: It would be good to add a statement on why heterotrophic N2 fixers improve the representation of deep oxygen as a key result of your paper.

**Response:** We now put more emphasis on oxygen in the discussion and added the following text:

"This model improves the representation of NO3 and O2 at depth (Figures 3, 7). Changes in deep NO3 can be explained through the enrichment of detritus, while changes in O2 occur because the heterotrophic group becomes an additional sink of O2 at depth."

P10, Line 18-20: Need to comment on potential reasons why H3 accumulates NO3 over time.

**Response:** We now state this more clearly:

"All models that consider nitrogen fixation accumulate nitrogen at different rates, as they enrich the nitrogen content of detritus, which is then remineralized at depth over time."

P11, Line 3-8: Need to add comments on why H3 and H2 have much higher N2 fixation rate than observations between 0-DCM in the Summer 2010.

**Response:** In the H3 and H2 models the large  $N_i$  fixation rates during summer are due to the contribution of blooming Trichodesmium. This is likely to be overestimated under the current model configuration and requires further calibration as more information becomes available to verify the seasonal cycle of nitrogen fixation in the region. We are acknowledging this by including the following sentence:

"... massive blooms are rare in the Gulf of Aqaba (Foster et al., 2009; Mackey et al., 2007) and the model probably overestimates the contribution of Trichodesmium spp.'s annual blooming to total  $N_2$  fixation rates, as seen in the much larger surface  $N_2$  fixation rates generated by H2 and H3."

P11, Line 17-18: I would be more subtle about the models' abilities to replicate N\* as it is still quite far from the observations (see my main comments above).

**Response:** Following this and other reviewer's comments we removed the figures and discussion concerning N\*.

P12, Lines 17-32: While this is an interesting section about the contribution of N2 fixation on PP, why include results from H3a which is not as realistic as H3?

**Response:** Following the Reviewer's recommendation, this text was removed.

P13, Section 5.3: One of the main points of the paper is to highlight the important role of heterotrophic nitrogen fixers. I feel this section could then be a lot stronger highlighting all the effect of heterotrophic N2 fixers on the ocean biogeochemistry of this region. Here for instance I would mention that heterotrophic N2 fixers improve the NO3 and O2 concentrations at depth, as well as the contribution of heterotrophic bacteria to total N2 fixation. Also, would it be possible to plot the model difference between H2 and H3 to show the impact of heterotrophic N2 fixation?

**Response:** We have modified the text accordingly. In term of plots, we have decided to show only the difference in terms of total  $N_2$  fixation, which presented in Figure 10b.

Figure 10 is not mentioned in the main text. It looks interesting so probably worse describing it at some point.

Response: Thank you for pointing this out. Figure 10 is now referred to in sections 5.2 and 5.3

Response to Comments by Reviewer 2 (Reviews are included in black font; Responses are in blue font)

**GENERIC COMMENT:**

This paper presents a model for heterotrophic nitrogen (N) fixation, implements it in a 1D context in the Arabian Sea and uses the model to test hypotheses on the relative contributions to N fixation by the different organisms. The subject fits perfectly in the Journal remits, and the work is highly relevant and it will be an important contribution to the topic of N fixation. I particularly liked the use of genetic algorithm for calibration and use of the model to test hypotheses. Authors set up a generally good framework to perform those test, unfortunately I believe that some further tests are needed in order to properly attribute the changes in the model outputs to the N fixation trait (see main comments).

**Response:** We are grateful for the positive assessment and appreciate the constructive comments. As described in our detailed responses below, we have modified the manuscript accordingly and believe it has improved significantly as a result.

MAIN COMMENTS: First of all, I would strongly encourage authors to be more comprehensive in the model description the supplement, because some key details are not clear, particularly on how the equations change in the different model set-up. In particular, in model H2 how the equation for zooplankton growth changes? Does Zooplankton see a single pool of phytoplankton formed by non-fixers and unicellular fixers, or does it graze separately on both? Given the non-linearity of the limitation function, the two options are very different. I would suggest writing explicitly all equations of H1, H2 and H3 that differ from H0 instead of summarising with sentences like "All other state variable equations are modified accordingly"

**Response:** In response to this suggestion we have now included the complete sets of equations in the supplements and have described them in more detail to clarify the model assumptions.

The main concern is that authors directly compare models with very different structure and then attribute all changes observed in the results to the process without separating the impact of the biogeochemical process from the impact of the different model structure. For instance, in H3 authors added the heterotrophic nitrogen fixers, by adding to the implicit first-order mineralisation scheme of the small detritus, a more dynamic one that includes an explicit heterotrophic group (Hf). Such a big change in the model structure is bound to profoundly impact the model results, regardless of the N fixation ability of the heterotrophic group, because the whole dynamic of mineralisation is changed. I would recommend the authors to implement a H3' model where a non-fixer group of heterotrophic organisms uses organic and inorganic for of both N and P is used as Hf.

The comparison between H0 and H3' would enable to understand how much of the mismatch between simulated and observed bottom waters N and O2 is due to an underestimation of mineralisation, while comparing H3' with H3 will allow to assess how the N fixation trait influence those dynamics. The comparison of H0 and H3' is much more important because the mineralisation

rates have not been calibrated, and therefore could be affected by an initial bias.

**Response:** We find this to be an excellent suggestion and have performed the additional simulation as suggested. In the model description we have added:

"An intermediate version H3' is used as a control, where the heterotrophic organisms do not fix nitrogen and are limited by the availability of nitrogen in inorganic forms and from small detritus. Model H3 eliminates the nitrogen limitation and the heterotrophic group becomes a heterotrophic diazotroph group."

The new results are included only in Figures 3 and 7, and described in section 4.2 "Model Results" to preserve the clarity of other plots. This additional simulation did not affect our main conclusions.

Similarly, when comparing model with 1 phytoplankton group (H0) with models with multiple PFT, all trophic dynamics can change, due to non-linearity in the graz- ing. Since Zooplankton dynamics are not shown, nor detailed equation for grazing and zooplankton growth in the different models, it is impossible to me to assess if the implementation of H1' and H2' similar to H3' are to be recommended or not.

**Response:** While we have tried to keep the model structures as consistent and comparable as possible, we acknowledge that comparing models with different complexities is challenging due to the various changes that may occur as consequence of changing trophic structure. We have expanded the model equations in the supplement to improve the clarity. However, further examination of the effects of different grazing functional forms, or other similar exercises are beyond the intended scope of the study.

Another main comment is related to the Redfieldian assumption. While I fully acknowledge the long tradition of Redfield ratio based models and data analyses, their power and their advantage, I'm always a bit concerned when these are used to draw conclusions on nutrient ratio dynamics, particularly in the short temporal and spatial scales. Phytoplankton internal nutrient ration and nutrient uptake are far from being constant and fixed to the Redfield ratio and they also varies a lot from species to species (e.g. Geider and La Roche, European Journal of Phycology, 2011, http://dx.doi.org/10.1017/S0967026201003456). I appreciate that this complexity is impossible to fully reproduce in biogeochemical model and therefore the Redfield assumption can still be used as first order approximation in simple biogeochemical model, however I would not use those model to analyse the instantaneous dynamic of nutrient ratios because this will be strongly affected by the

huge assumption of fixed stoichiometry. Figure 6 itself shows how the model is not able to capture the wide variability of DIN:DIP ratio. For this reason, I would suggest to cut the part related to N\*, or alternatively, repeat the analysis using annual means of DIN and DIP and include a discussion on the importance of non-Redfieldian dynamics.

**Response:** Following this and other Reviewers' comments we have decided to remove figures and discussion concerning N\*. Also, sensitivity experiments to N:P ratios now illustrate the effect of changes in non-fixing phytoplankton and diazotroph stoichiometry (see responses to Reviewer 3).

SPECIFIC COMMENTS: Page 5, lines 19-20: in H0' N fixation and denitrification are balanced: where did denitrification occur? In the benthos? I recommend adding some detail to better interpret the vertical dynamics simulated by this model implementation

**Response:** The denitrification representation was included in the model description in the supplement: "When present, the denitrification flux follows Fennel et al. (2013) with a loss fraction of 6 mol  $N_2$  per mol of organic matter remineralized". We added this descriptive sentence in the methods section of the main text.

Section 4.2.2.: Top left panel of figure 5 shows that model H2 and H3 are significantly overestimating surface nitrogen in the last 4 years of calibration, with the exception of the deep mixing events in winter 2007/2008. H3 largely overestimates surface nitrogen also in the validation period (figure 8). This important dynamic is not discussed in the paper.

**Response:** Geomorphology and bathymetry limit water exchange fluxes between the Gulf of Aqaba and the Red Sea to the upper 300 m. It this unlikely that horizontal transport could explain the observed accumulation in deep NO3, but exchange of surface waters during summer months does occur. A possible explanation is that some of the DIN produced should be exported to the outside of the Gulf, which our 1D model does not account for. The following sentence was added to section 5.4:

"It is, therefore, unlikely that horizontal transport could explain the observed accumulation of deep NO3. Nevertheless, transport of nitrogen-enriched sub-surface waters from the Gulf of Aqaba towards the exterior may dampen the long-term accumulation of nitrogen."

Section 4.2.4.: while I agree that H3 better compares with observed deep values, in the last couple of years a significant trend in deep nitrogen appears in the simulation and it's not in the data. I

suggest authors to comment on that.

**Response: Please see previous response concerning export of DIN to outside the Gulf.**

TECHNICAL COMMENT: Page 12, line 18: in 10b, the dot corresponding to Capone and Carpenter 1982 shows a N fixation equal or close to 0, that is quite different from the values simulated by the different flavour of H3 Figures 2,4,7: I recommend the authors to redraw the picture using a perceptually uniform and colour-blind friendly colourmap like viridis, inferno, magma or plasma in Python or Parula in Matlab. More details on the importance of this in the following video https://www.youtube.com/watch?v=xAoljeRJ3lU

**Response:** Taken, we have redrawn the data figure in the color-blind friendly colormap "parula".

**Response to Comments by Reviewer 3**

(Reviews are included in black font; Responses are in blue font)

The authors present a study which compares a series of 1-D biogeochemical models of increasing complexity with respect to the representation of different diazotrophic organisms against an in situ time series data set from a location in the Gulf of Aqaba in the Red Sea. Comparison of the output of these models with the in situ data and specifically the nutrient concentrations and ratios/differences between N and P con- centrations (as quantified by the derived N\* variable) is subsequently used to argue for a substantive contribution by heterotrophic diazotrophs to N2 fixation within the region. The manuscript is well written and in general the study rationale and model experiments appeared well designed (although see specific comments below). The subsequent results and potential implications of the study were certainly interesting and overall I felt there was much of value within the study. However, as outlined below, I have a few concerns I would like to see the reviewers address.

**Response:** We are grateful for the overall positive assessment and appreciate the constructive comments, which we respond to in more detail below. We believe the resulting edits have greatly improved the manuscript.

**Major comment:**

The authors appear to undertake a thorough job of optimising many of the parameters related to their model (see e.g. Page 6 and Table 2). However, given the key question(s) being addressed

within the study I was somewhat surprised that potential variability in what I would consider to be the key parameters in dictating how the diazotrophs interact with the N and P cycles were fixed, with no exploration of potential variability in these parameters. Specifically, the values of the N:P ratio within both the non-diazotrophs and the diazotrophs were fixed at 16:1 (Page 1 of Supplm.) and 45:1 (Page 5 of Supplm.) respectively. In contrast it is now fairly well recognised not only that N:P ratios within organic material can vary (see e.g. Martiny et al. 2013 Nature Geo. 6 279-283) but also (and crucially within the current context), that inferences of N2 fixation rates and interactions between diazotrophy and the cycling of N and P are highly dependent on both assumed values of these ratios and any variability within these (Mills and Arrigo 2010 Nature Geo. 4 412-416; Weber and Deutsch 2012 Nature 489 419-422). Consequently, I would suggest the authors should at least consider the implications of their assumed fixed N:P ratios for their interpretation and conclusions and perhaps also consider performing some sensitivity analysis around these currently fixed assumptions.

**Response:** This is a very valuable point. We performed a sensitivity analysis in which we have increased and decreased the fixed N:P ratios in the non-fixing and N fixing organisms. Section 4.2.3. and figures 5 and 6 address these experiments. Overall, changes in N:P ratios affect PO4 concentrations more strongly than NO5 concentrations.

**Specific comments:**

Page 3, Line 2: It would be worthwhile directly stating the laboratory based studies considered here were specific to Trichodesmium. As far as I am aware we have little information on how other groups might be expected to respond to the drivers men- tioned.

**Response:** Agree, the laboratory experiments cited consider only *Trichodesmium*, while the model study of Dutkiewicz et al. (2015) considers a set of generic diazotrophic organisms various sizes and growth rates. The following line was added:

"To our knowledge, these laboratory experiments have only explored the reaction of Trichodesmium and less information is available about the effects of climate trends on other diazotrophic organisms."

Page 5, Line 21: '... a generic autotrophic diazotroph...'

**Response:** Thank you for pointing out this typo.

Page 6, Lines 10-25: I was unclear whether this parameter optimisation method was performed for each of the models (H0 – H3 etc) independently or a single parameter set was used? Additionally, see major comment above, did the authors consider using the parameter optimization method for the non-diazoptroph and diazotroph N:P ratios? See also Page 14, Lines 11-16, I was unsure why this choice was made, it appears to be a big assumption within the current context.

**Response:** The optimization was carried out independently for each of the models. We clarified by modifying the introductory statement:

"Parameter optimization refers to the minimization of misfit between model and observations by adjusting model parameters. We applied the method first to systematically calibrate the most sensitive parameters of H0 and then to independently re-calibrate parameters in H1 to H3 after the introduction of diazotrophs."

Page 7, Line 13: maximum reported growth rates for Trichodesmium are actually >0.5 d-1, see Hong et al. (2017) Science 356 527-531

**Response:** Thank you for pointing us to this recent reference. We added it and modified the text accordingly in the revised manuscript.

Page 8, Line 9: see major comment above. Either this assumption should be justified, or, preferably I would suggest, some effort could be made to perform a sensitivity analysis of how the assumption influences the results/conclusions.

**Response:** Agree, see response above.**

Page 8, Line 14: again related to comments above, some speculation on how this happens would seem appropriate in the context of this study. As a suggestion, uptake at high N:P ratios by non-diazotrophs might be one potential mechanism for shifting from inputs of nutrients with an apparent 'excess' N (i.e. positive N\*) to an apparent deficit (negative N\*), see e.g. Mills and Arrigo (2010).

**Response:** Following reviewers suggestions, we have removed N\* figures as well as the text describing it. This line is no longer in the text.

Page 9, Line 19 (also Page 12, Lines 7-8): it is notable that even the most complex model struggles to reproduce the observed range in N\* and I wondered whether the restriction placed on the models through the assumption of the fixed N:P ratios may be responsible for this?

**Response:** That is indeed a possibility that we had not explored. As brought up by Reviewer 1, other alternatives include horizontal physical transport in and out of the domain, phytoplankton stoichiometry, atmospheric deposition, and preferential PO4 remineralization. The latter may particularly affect PO4 at depth, while transport may affect surface values during summer (when exchange of surface waters with exterior waters has been reported to occur). We have extended our discussion about limitations and uncertainties to acknowledge these considerations. Nevertheless, given the reservations about our discussion of N\* that were expressed by several of the Reviewers, we have decided to remove this figure and other references to N\* in the text.

A final general point which is also related to many of those above, within the context of this study I felt that some of the important details relating to the model which were presented within the supplement would be more appropriately outlined within the main body of the text as they are likely fundamental to interpretation.

**Response: The following text was added to the description of the base model to improve clarity:**

"H0 is the base model without diazotrophic plankton groups and follows the model equations described in Fennel et al., (2006, 2013). In general, the model follows Monod kinetics using a fixed N:P ratio ( $R_{N:P}^{nf}$ =16) for phytoplankton and zooplankton. We test the H0 model with and without a sediment denitrification flux (model versions H0 and H0', respectively). H0 includes denitrification but no N2 fixation, as no diazotrophs are considered. H0' does include neither denitrification, nor diazotrophic organisms, and thus the underlying assumption of this model version is that there is a balance between inputs from N2 fixation and losses of fixed nitrogen due to denitrification. When present, the denitrification flux follows Fennel et al. (2013) with a loss fraction 6 mol N2 per mol of organic matter remineralized at the sediment-water interface. This results in an average sediment denitrification flux of 0.25 ± 0.46 mmol N m2d4, with a maximum value of 3.01 mmol N m2d4." As requested by Reviewer 2, we also expanded the supplement to include all model equations.

**Response to Comments by Reviewer 4**

(Reviews are included in black font; Responses are in blue font)

**General evaluation**

This ms reports a 1D biogeochemical model analysis of time-series data from the Gulf of Aqaba from 2006–2014. The authors compare the behaviour of models with different diazotroph community structures representing various combinations of autotrophic and heterotrophic

diazotrophs. While all model versions perform similarly with respect to surface chlorophyll, only models with diazotrophy can reproduce observed nutrient (N:P) ratios and heterotrophic diazotrophy is required to explain the vertical structure of nutrient and O2 concentrations.

In general, I find this study somewhat unconvincing. The model is overly simplistic in its mechanistic foundation and ignores processes I consider essential for this kind of analysis. While I do not dispute the potential importance of heterotrophic diazotrophy for marine biogeochemistry, the conclusions and particularly the title appear overly optimistic and not well justified. The ms also appears to have been prepared rather sloppily and not thought through. The main problem is that all diazotroph parameters are unconstrained by the data, which, as outlined below, may be a consequence of the overly simplistic nature of the model or of an inappropriate cost function. Thus, in order to turn this ms into a useful contribution, the model or the cost function (or both) must be redesigned so as to achieve sensitivity to the diazotroph-related parameters.

**Response:** We respond one-by-one to each of the critical points that the Reviewer raises:**

With regard to the comment that "the model is overly simplistic," we would like to state that our analysis is exploratory and focused on the influence of  $N_2$  fixation, hence complexity in other parts of the ecosystem, although necessary in any global application, is intentionally minimized here. The Gulf of Aqaba is an oligotrophic system and our relatively simple model structure is able to capture the observed variability.

With regard to the title being "overly optimistic and not well justified," we have changed the title according to this Reviewer's suggestion (see response to detailed comment below).

With regard to the manuscript being "prepared rather sloppily and not thought through," we would like to point out that some of the information the Reviewer is requesting (i.e. literature sources of parameter values) is actually provided in the manuscript (Tables 2 and 3) and discussed in the text for diazotrophic organisms (section 3.3.2). In our revision we incorporated the additional information that the Reviewer requested, i.e. more details on the sensitivity experiments in the supplement, and initial conditions, more details on the ranges and literature sources for diazotrophic organisms in the main text (see detailed comments below).

With regard to "diazotroph parameters are unconstrained by the data," we would like to point out that this is not an issue with the model or the cost function. It is generally accepted among modellers that measured bulk properties like chlorophyll, nutrients and oxygen do not constrain most rates including rates of grazing, phytoplankton mortality and, in our case,  $N_2$  fixation (see e.g. Ward et al., 2010). No redesign of the cost function or the model will change the fact that the measured properties in the Gulf of Aqaba do not directly constrain  $N_2$  fixation rates, simply because that information is not contained in the observations. The observations capture the impact that  $N_2$  fixation may have only indirectly through its influence on deep-water nutrient ratios.

Ward, B.A., Friedrichs, M.A.M., Anderson, T.R., Oschlies, A., 2010. Parameter optimization techniques and the problem of underdetermination in marine biogeochemical models. J. Mar. Syst. 81, 34–43.

**Specific points**

1. Starting with the title, I find the wording inappropriate. While it might be possible to obtain biogeochemical evidence from a model analysis, this is certainly not the case here. I would suggest something like "Modelling heterotrophic N2 fixation ..."

**Response:** We have changed the title as follows:**

"Modelling the biogeochemical effects of heterotrophic and autotrophic N2 fixation in the Gulf of Aqaba"

2. Model structure. Although the authors stress that they intended to analyse mechanistic assumptions (l. 15, p. 14), I find that the model is mechanistically rather weakly founded. While simplicity is of course an important goal in model development, one must take care not to over-simplify and neglect essential processes. I think this should be at least discussed thoroughly to put the results into the right perspective. The two assumptions I find most troubling are those of (1) constant (Redfield) stoichiometry of the autotrophs and (2) obligate diazotrophy, both of which are mechanistically wrong. Fernandez-Castro et al., J. Plank. Res. 38:946 (2016), FC in the following, applied a model with variable stoichiometry and facultative diazotrophy in the subtropical North Atlantic, where the vertical distribution of N, P, and N\* poses similar difficulties as in the present ms. The model of FC is otherwise very similar in structure to the present one (phytoplankton, diazotrophs, zooplankton, detritus, nutrients, DOM), so I think the differences should be discussed, particularly with respect to the relations among stoi- chiometry, export and remineralisation.

**Response:** We have included the citation to FC in the revised manuscript in the Introduction and

the Discussion. With regard to the particular assumptions mentioned by the Reviewer, namely constant (Redfield) stoichiometry of the autotrophs and obligate diazotrophy, we would like to comment that the overwhelming majority of models (including those used in IPCC projections) assume constant stoichiometry and, where diazotrophy is explicitly included, obligate diazotrophy. In our revision, following another Reviewer suggestion, we included an additional sensitivity experiment to show the implications of changing the stoichiometry of the diazotrophs and non-fixers. The question of obligate diazotrophy may be boil down to semantics because autotrophic diazotrophs that aren't fixing N, would behave like a non-diazotrophic phytoplankton, which are included in our model. We would also like to comment that our focus is not on the physiology of diazotrophs, which is analyzed in more detail by Fernandez-Castro et al. and citations therein. We acknowledge that when modeling diazotrophic cells, N allocation mechanisms are important to understand how diazotrophs are able to fixate N, under conditions that are traditionally thought to limit the process. However, our focus is to approach the assumptions about diazotrophs niches with a trait-based perspective. We have also changed the use of "mechanistic model" to "trait-based models" for clarity.

Comparing the parameter settings between FC and the present model, I notice a very strong discrepancy (more than a factor of 10) in the initial-slope parameter (alpha) for photosynthesis in diazotrophs, although the units are the same in both models. It is not clear from the ms how or why the very low alpha was chosen (no reference given and not optimised). But it appears to be an important parameter given that the analysis is about the vertical structure and alpha basically defines how deep in the water column autotrophic N2 fixation can occur.

**Response:** Our values of alpha are within the range of values typically used in ecosystem models with similar formulations (see Doney et al. 1996; Fennel et al., 2001; Fennel et al., 2002; Schartau and Oschlies 2003; Fennel et al., 2006; Moore et al., 2004). Please also notice that while FC's and our alpha parameters share the same notation, they may not refer to the same parameter because our light limitation formulations are not the same. It is not straightforward to compare these values directly to each other. Also, we would also like to note that FC's alpha values were subjectively adjusted from the original model configuration by Pahlow et al., 2013, but not optimized. FC report these parameters were adjusted to reduce the depth of N2 fixation in their model results and even then, they obtain significant differences between the simulated and observed vertical structure of N2 fixation. To us, these differences in parameter values simply exemplify the fact that transferring parameters measured in laboratory cultures to mathematical models representing the real ocean or in-between different models is challenging and corresponds to a different discussion in itself. Table

3 now emphasizes these references, which were only included in the text previously.

Doney, S., D. M. Glover, R. G., Najjar, 1996. A new coupled, one-dimensional biological-physical model for the upper ocean: Applications to the JGOFS Bermuda Atlantic Time- series Study (BATS) site. Deep-Sea Research II, Vol 43, No. 2-3 pp. 591-624.

Fennel, K., M. Losch, J. Schröter and M. Wenzel, 2001. Testing a marine ecosystem model: Sensitivity analysis and parameter optimization. *Journal of Marine Systems* 28/1-2, p.45-63

Fennel, K., Spitz, Y.H., Letelier, R., Abbott, M.R., 2002. A deterministic model for N2 fixation at stn. ALOHA in the subtropical North Pacific Ocean. Deep-Sea Res. II 49, 149–174.

Fennel, K., Wilkin, J., Levin, J., Moisan, J., O'Reilly, J.E., Haidvogel, D., 2006. Nitrogen cycling in the Middle Atlantic Bight: Results from a three-dimensional model and implications for the North Atlantic nitrogen budget. Glob. Biogeochem. Cycles 20, 14. doi:10.1029/2005GB00245.

Schartau, M., Oschlies, A., 2003. Simultaneous data-based optimization of a 1D-ecosystem model at three locations in the North Atlantic: Part I - Method and parameter estimates. J. Mar. Res. 61, 765–793.

Moore, J.K., Doney, S., Lindsay, K., 2004. Upper ocean ecosystem dynamics and iron cycling in a global three-dimensional model. Glob. Biogeochem. Cycles 18(4).

Another parameter that appears rather low is the maximum growth rate of the au- totrophic diazotrophs. For example, Holl & Montoya, J. Phycol. 44:929 (2008) re- ported growth rates greater than 0.6/d for Trichodesmium grown in a chemostat, so a maximum (actually potential) rate parameter of 0.25/d appears unrealistically low. My impression is that these low settings reduce diazotrophy too much, maybe just compensating for the assumption of obligate diazotrophy but maybe also being responsible for the requirement of aphotic N2 fixation in the present model.

**Response:** Please note that the maximum growth rate in the model is temperature dependent and that 0.25/d is the reference value at 0oC (as stated in Table 3). At typical water temperatures in the Gulf of Aqaba of 20oC this results in an actual maximum growth rate of 0.97/d, close to the value mentioned by the Reviewer.

Further, the authors say that the diazotroph parameters were unconstrained by the data and that the parameter setting were taken from the literature, but do not provide references in Table 3 or elsewhere. The ms also does not say how it was determined that the parameters were unconstrained by the data. This seems inappropriate to me, since this is specifically a model study about diazotrophy, so I expect that great care is taken to select appropriate parameter settings. The fact that the diazotroph parameters are unconstrained by the data makes the choice of data appear questionable to me. In my view, the data should be able to constrain the most important aspects of a model's performance, and if this is not the case, one should try to either find better data or develop a better cost function (see below). The problem is that the inability to constrain the model parameters with the data implies that the associated processes are actually irrelevant. The simple fact that the authors observe better model performance when including diazotrophs implies that the associated parameters.

**Response:** With regard to the Reviewer's assertion that "parameter setting were taken from the literature, but do not provide references in Table 3 or elsewhere," we would like to point to Table 2 which lists ranges for each parameter from the published literature with the corresponding references (see columns 3 and 6). Table 3 now emphasizes these references too, only included in the text previously.

In response to the comment that we do "not say how it was determined that the parameters were unconstrained by the data": Showing that measurements of chlorophyll, nutrients and oxygen do not constrain N2 fixation rates is not within the intended scope of this paper. In fact, it appears self-evident to us. What we here mean by "unconstrained" is that, for example, chlorophyll alone cannot provide any information about how much chlorophyll is due to diazotrophs and how much is due to non-diazotrophs, therefore this variable alone is unable to help in the determination of diazotrophic parameters. The systematic calibration method we use relies on having direct observational counterparts (i.e., from the same location at least) to the model output. From previous knowledge and experience using optimization methods, we would need comprehensive N2 fixation and/or size-structured diazotrophic biomass data in order to constrain diazotrophs parameters.

With regard to the comment that "the choice of data appear questionable to" the Reviewer, we would like to respond that we used all the data that was available. The suite of available measurements is very typical of multi-year oceanographic time series (most aren't as comprehensive and well-funded as the HOT and BATS programs).

**With regard to the cost function we refer to the last paragraph in our response to the first comment.**

3. Model evaluation. The authors report that they performed sensitivity analyses to obtain information of sensitive model parameters but they do not say how the sensitivity was quantified nor present any results from the sensitivity analyses. This could well be done in the supplement, but it is important for those who want to work with the model later.

**Response:** The information about the preliminary sensitivity analysis in now included in the Supplement.**

The authors mention that they considered the first year of the model simulations as spinup but do not say how the model was initialised (from observations? what about the non-observed variables?). From my own experience with 1D modelling, one year is a rather short period for a spinup. Did the authors try longer spinups in order to find out whether the model is sufficiently close to a quasi-steady-state after one year? This should be discussed as well. It is this kind of omission, together with missing entries in the list of references (e.g., Fernandez 2011 and Smith 1936), that leaves an impression of sloppiness.

**Response:** We have included this information in the methods section as follows:

"NO3, NH4 and PO4 initial total concentrations match the observed total inventories, using homogenous concentrations throughout the water column of 2.5 mmol m-3, 0.05 mmol m3 and 0.15 mmol m3, respectively. Vertical nutrient concentrations are redistributed within few months and replicate the observed vertical distributions well starting from October 2005. Non-fixing phytoplankton, zooplankton and detritus are initially set to a homogeneous small value of 0.1 mmol N m3, which also readjust rapidly because the adjustment timescales for these variables are short (Fennel et al., 2006). Diazotrophs initial values are set to lower densities than non-fixing organisms, with a homogenous total value of 0.03 mmol N m3 (i.e., models with multiple diazotrophs maintain the same amount of initial diazotrophic biomass)."

4. Parameter estimation. The authors apply RMSEs of absolute concentrations to obtain a measure of model-data misfit. This cost function will not be sensitive to large relative deviations if the absolute concentrations are low. Thus, it is only logical that the inability of the model to reproduce the negative N\* in the surface waters "is not a source of large data-model discrepancies" (1. 8, p. 12). Introducing relative-error information or local scaling into the cost function could help here. The most important shortcoming of the authors' cost function, however, is that it neglects error

correlations, see, e.g., Schartau et al., Biogeosci. 14:1647 (2017).

**Response:** Our cost function does indeed scale depending on the data type by weighing the contributions from the different variables by their standard deviations.

5. Figures. The use of log-scales in Fig. 9 makes it impossible to see the differences among models and between models and data. Please use a linear scale.

**Response:** We chose the log-scale to be able to visualize differences at very low values, common for these rates in the region.

6. Conclusions. As it stands, the conclusions are not sufficiently supported be the model analysis described. In particular, the conclusions about aphotic N2 fixation are compromised by the choice of unrealistic parameter values constraining autotrophic diazotrophy to the very surface. If inferences about heterotrophic diazotrophy are to be drawn, at least the parameters determining the depth distribution of autotrophic diazotrophy must be analysed with a detailed sensitivity analysis. The current analysis cannot say whether the deep N signal is really due to aphotic N2 fixation or exported material from the surface.

**Response:** As per our responses above, the parameters for autotrophic diazotrophs are fully consistent with the existing literature.

**2. SUMMARIZED LIST OF CHANGES**

- We changed our manuscript title to better reflect the scope of our results.
- We changed our focus on N\* and removed plots and discussion concerning this tracer, following reviewers suggestions.
- We performed an additional model experiment (H3'), which served as a control of the effect of adding a non-fixing heterotrophic organism. Vertical distributions of nitrate, phosphate, chlorophyll and oxygen results are shown. Our main study conclusions were not affected.
- We included a sensitivity analysis to changes in the N:P constant ratios of N2 fixing and non-fixing organisms. Our main study conclusions were not affected.
- We extended the discussion, in particular the section concerning limitations and uncertainties. This was done to include relevant studies suggested by our reviewers.

- We extended the supplement to include a more detailed description of the equations of all model versions.
- We extended the supplement to include a description of the preliminary parameter sensitivity analysis cited in the text.
- We clarified relevant details in our methods, such as how denitrification is implemented in the models.
- We changed the use of "mechanistic models" to "trait-based models."
- We specified references followed for diazotrophic parameters in the corresponding table. Previously these were cited only in the text.
- We changed the colorscale of results plots to a colorblind friendly colormap.

**3. MARKED-UP MANUSCRIPT VERSION**

In yellow sections that have been modified.

[revised manuscript text omitted]

---

## Author Response (AR2)

**Response to reviewer**

We appreciate the time this reviewer has taken to comment on our manuscript for the second time. We have considered their suggestions and answer them in detail below.

[reviewer comments in blue]

The authors have done a reasonable job in responding to my previous review. However, I do have a couple of concerns related to two of the more substantive changes which the authors implemented in response to comments previously raised by both myself and the other reviewers.

Major comments

Firstly, many of the reviewers, myself included, commented on the poor agreement between the modelled and the observed N* distributions in the previous version. In response to this the authors have removed direct comparison between observed and modelled N* and instead rely on comparisons between model results and observations of N and P separately. However, the use of N* (or effectively N:P ratios), which will reflect the decoupling between the cycling of N and P (see e.g. Sarmiento and Gruber 1997 and numerous subsequent usage), would still seem like the most obvious and direct constraint on the N cycle in general and N2 fixation in particular within the model. As such I personally found the decision to remove this comparison, rather than, for example, to directly address the prior weakness by investigating how the added sensitivity analyses (see below) influenced model reproduction of N* (and/or N:P ratios) to be a somewhat unsatisfactory response.

Response: The decision to remove N* figures and discussion was taken after the explicit suggestion of one of our first-round reviewers. Also, in response to those comments, we had added significant discussion about limitations and additional sources of N and P not included in the model, which could improve model performance. In this new version, we have reintroduced the observed and simulated N* figures and accompanying discussion. However, we feel that the addition of further figures analyzing the sensitivity of N* to planktonic N:P ratios does not add significant information—to some extent, this information can be inferred from the effects on N and P. As N:P ratios mainly affect P concentrations, we would expect a deviation of N* toward excess phosphate when phosphate is increased (Fig 7 b, h in the revised manuscript) and an enhancement of excess nitrate when phosphate decreases (Fig 7 d in the revised manuscript).

Additionally, I was pleased to see the authors perform a sensitivity analysis investigating how changes in the assumed ratios of N:P within both non-N2 fixing phytoplankton and nitrogen fixers influenced their results. However, this sensitivity analysis appeared quite limited. In particular, I was unclear why this analysis was only performed for the H1 version of the model? Also, I note that the sensitivity analysis did not really cover the parameter space of different assumed N:P ratios for phytoplankton and nitrogen fixers very thoroughly, as they were varied independently around their initial assumed values. Wouldn't it be more appropriate to perform

a sensitivity analysis over a matrix of values fully investigating the parameter space?

As an example of how the two points above might interact, I would particularly have liked to see whether a sensitivity analysis of the assumed N:P ratios within the H3 version of the model might influence the ability of this most complex structure to simultaneously reproduce P and N observations, as indicated by N* values (or maybe simply N:P ratios in the subsurface).

Response: We acknowledge that investigation of the effects of planktonic stoichiometry is relevant for a better understanding of N* variability. Nevertheless, an extensive exploration of the model sensitivity to N:P is beyond of the intended scope of the present manuscript and may distract the narrative from our main arguments. One-at-a-time variations in parameter values are a frequently used exploratory tool and we have used sensible value ranges for both fixing and non-fixing phytoplankton types (10-28 and 19 – 59, respectively). In the manuscript, we only show a few examples of the results (both increasing and decreasing the initial N:P value), but 6 different values within those ranges were tested generating similar results. While N:P ratios may vary temporarily outside the value ranges we tested, using those values as fixed constant ratios would generate unrealistic responses.

More sophisticated methods to investigate sensitivity are often the subject of independent analysis. They may involve techniques to optimally sample the parameter space and/or the use of statistical model emulators to minimize computational time required by such extensive evaluations of the parameter space. We estimate that a minimum of ~625 model runs would be required for the sensitivity analysis suggested by the reviewer.

Additional more minor comments

Page 4, Line 20: this statement could perhaps be more specific given the different perturbations/versions of each of the models (H0-H3) now investigated.

Response: We modified the text from *"We analysed the role of autotrophic and heterotrophic N$_2$-fixing organisms in determining biogeochemical patterns at an open pelagic site (Station A), located in the northern Gulf of Aqaba, by testing **four alternative ecosystem** model versions"* to *"We analysed the role of autotrophic and heterotrophic N$_2$-fixing organisms in determining biogeochemical patterns at an open pelagic site (Station A), located in the northern Gulf of Aqaba, by testing **four main alternative ecosystem model versions and six model subversions with minor variations**"*

Response: Thank you, corrected.

Response: We applied the optimization method to parameters that often exhibit large value ranges, as an alternative to calibrate the model subjectively. While we acknowledge that N:P

ratios can be variable, there is more consensus about N:P ratios for N2 fixing and non-fixing phytoplankton when used as fixed values in models. Here we kept these values to maintain consistency with common assumptions in previous models. The method could be applied to N:P ratios; however, in order to obtain more reliable solutions to these parameters, PON and POP data would be needed.

Page 9, Line 25 onwards. Related to second major comment above, in the absence of any further analysis, some rationale for why this sensitivity analysis was only performed for H1 and only around the initially assumed values (rather than more fully exploring the parameter space), should be provided here.

Response: As all diazotrophic groups share some characteristics of the phosphate uptake parameterizations, the behaviour of the generic diazotroph is indicative of potential effects in the most extensive model versions in a simplified context. We have added this comment on the main text.

---

## Author Response (AR3)

Dear Dr. Jack Middelburg,

Thank you handling the edition of our manuscript to a success. We think that the manuscript has been greatly improved throughout the revision process. We also appreciate the considerable time you have taken to carefully read our manuscript. Thanks to that, we have corrected all the typos and references we previously missed. Below is a detailed description of the latest changes we implemented:

- P. 2, l. 16: correlation between heterotrophic bacterial productivity and N2 fixation
**Response: we implemented your suggestion and added the word "heterotrophic" to this line.**

- P. 4, l. 10: NO3 and PO4 reach 2 and 0.1 microM, respectively…. (it appears that you have them swapped in the text when comparing with the figure, p. 5 information and my understanding of the system). Please check.
**Response: thank you for noticing this typo, the values were swapped indeed. We corrected it.**

- P.5 , l. 20, l. 23 and all through text: I propose to use the term non-N2-fixing rather than non-fixing organisms/phytoplankton because phytoplankton fixes CO2.
**Response: we implemented this suggestion throughout the text and in the description of parameters in the tables.**

- Check your reference list: e.g. Benavides et al. 2017 is not in it.
**Response: thank you for pointing this out. We revised the reference list thoroughly for its completeness. References cited in the supplements are now also included in this list.**

- P 10, l. 4 and all through: check the use of superscript/subscript with NO3, PO4
**Response: Thank you for noticing this type. We checked the entire document and corrected it where necessary.**

- P.16, l. 21: which are the typically available ones in ..
**Response: the sentence was modified as suggested.**

- P. 16, l. 22: try to avoid the word experimental testing in a modelling study. Why not use the word explorative or alike? Experimental testing should be reserved for real experiments not numerical experiments.
**Response: We changed the word to "explorative" in this line and modified other similar instances throughout the text.**

Best regards,

Angela Kuhn